



Effect of Caribbean Water Incursion into the Gulf of Mexico derived from Absolute Dynamic
Topography, Satellite Data, and Remotely - sensed Chlorophyll-*a*
*Authors:*
**Juan Antonio Delgado[1,2,5]; Joël Sudre [3], Sorayda Tanahara[1];  Ivonne Montes[4],**
**José Martín Hernández-Ayón[5], Alberto Zirino[6]**
Author affiliations:
[1]*Facultad de Ciencias Marinas, Universidad Autónoma de Baja California, Transpeninsular*
*Tijuana-Ensenada, no. 3917, Fraccionamiento Playitas, CP 22860. Ensenada, Baja California,*
*México.*
[2]*Instituto Tecnológico de Guaymas/ Tec. Nacional de México, Guaymas, Sonora, México.*
[3]*LEGOS, CNRS/IRD/UPS/CNES UMR 5566, 18 av. Ed Belin, 31401 Toulouse Cedex 9, France*
[4]*Insitituto Geofísico del Perú. Lima, Perú.*
[5]*Instituto de Investigaciones Oceanológicas, Universidad Autónoma de Baja California,*
*Transpeninsular Tijuana-Ensenada, no. 3917, Fraccionamiento Playitas, CP 22860. Ensenada,*
*Baja California, México.*
[6]*Scripps Institution of Oceanography, University of California, San Diego, 9500 Gilman Drive,*
*La Jolla, California 92093, USA*
Corresponding author: Sorayda Tanahara (stanahara@uabc.edu.mx)
*Facultad de Ciencias Marinas*
*Universidad Autónoma de Baja California*

**Key points:**
Twenty-five years of satellite observations of absolute dynamic topography confirm the patterns
of Caribbean water intrusion in the Gulf of Mexico.
Larger volumes of oligotrophic waters from Caribbean Sea are entering the western Gulf of
Mexico and lowering the surface and near surface *Chl-a* concentration.


**Abstract**
The dynamics of the Loop Current (LC) and the detached Loop Current eddies (LCE's) dominate
the Gulf of Mexico's (GoM) surface circulation and transport Caribbean water (CW) into the
GoM. In this work, 25-years (1993-2017) of daily satellite data are used to investigate the
variability of these physical processes and their effect on chlorophyll-a (*Chl-a*) concentrations
from 1998-2017 including temporal changes, mean differences, and regional concentration
tendencies. Physical variables analyzed are absolute dynamic topography (ADT), oceanic currents,
and wind stress. From the ADT and oceanic current monthly climatologies, it is shown that there
is an annual intrusion of the CW with an inward incursion that starts in spring, peaks in the summer
(reaching to 26.58°N and 88.32°W) and then retreats in winter. Minimum surface *Chl-*
*a* concentrations ($<0.08$ *mg m$^{-3}$*) are found during the summer-autumn period inside the region of
maximum incursion of the CW; the opposite is observed during the winter period when the *Chl-a*
concentrations were at a maximum, e.g., $>0.14$ *mg m$^{-3}$*. The three-year running averages of ADT
40-cm isoline reproduce qualitatively the climatological pattern of 25 years showing that before
2002 the CW was less intrusive. This suggests that from 2003 onward, larger volumes of
oligotrophic waters from Caribbean Sea have invaded the western GoM and reduced mean surface
*Chl-a* concentrations. A direct comparison between the 1998-2002 and 2009-2014 periods
indicates that, in the latter time interval, *Chl-a* concentration above waters deeper than 250 *m* has
decreased significantly.
**1. Introduction**
The effects of global warming on the circulation of the world's oceans and its concomitant
consequences on the oceans' biological productivity are some of the most important scientific and



economic issues of our times. Forecasting of the effects of global warming on the ocean's resources
depends on having a clear understanding of the manner in which physical processes (e.g., solar
radiation, winds, ocean circulation and vertical mixing) affect primary production.    This
understanding is aided by the availability of remote sensing observations, unparalleled in their
spatial and temporal coverage of the earth's surface.  Since 1990, satellite data of   absolute
dynamic heights (ADT), chlorophyll concentration, and derived products (eddy kinetic energy
(EKE), geostrophic and Ekman currents) have been available to study the Gulf of Mexico (GoM),
an important socio-economic region for fisheries, petroleum, natural gas, and tourism. We have
availed ourselves of a 25-year time series of satellite data to study the relationship between the
physical dynamics of the GoM and its effect on primary production in the context of a global
warming scenario. Unlike previous studies, this work is based on detailed observations and
analysis of both the Loop Current (LC) and LC eddies (LCE), dominant features of the surface
circulation that transports CW into the GoM (Nowlin & McLellan, 1967; Tanahara, 2004; Schmitz,
2005). The LC in the eastern GoM is part of the North Atlantic Ocean Subtropical Gyre, an
essential contributor to the inter-hemispheric Meridional Overturning Cell (Schmitz & McCartney,
1993; Candela *et al.*, 2003; Schmitz *et al.,* 2005). This current carries warm waters from the gulf
to the North Atlantic through the Florida Straits via the Gulf Stream (Hurlburt & Thompson, 1980),
thereby also being an important contributor to the upper ocean heat budget of GoM (Liu *et al*.,

74    2012).

Within the GoM, Caribbean water enters the gulf through the Yucatan Channel to form the LC,
acting as the primary forcing mechanism (PFM) of this current. The current penetrates into the
gulf, reaching 28°N, near the Mississippi Delta. As it extends to the north, it forms a loop (Austin,
1955) that turns southeast to ultimately exit into the Atlantic Ocean.



Knowledge oft how the thrust of Caribbean water affects the LC is based on hydrographic data
(Leipper, 1970; Niiler 1976; Molinari *et al*., 1977; Behringer *et al.,* 1977; Huh *et al.,* 1981;
Paluszkiewicz *et al.,* 1983), remote sensing observations (Leben, 2005; Leben & Born, 1993;
Vukovich, 1988; Vukovich *et al.*1979), and, in the last twenty years, by numerical modeling
(Hurlburt & Thompson, 1980; Candela *et al.,* 2003; Oey *et al.,* 2005; Counillon & Bertino, 2009;
Sturges & Lugo-Fernandez, 2005; Wei *et al*., 2016; Cardona & Bracco, 2016). More recently,
novel developments based on artificial neural networks and empirical orthogonal function analyses
have also been applied to predict LC variation (Zeng *et al.,* 2015) effecting reliable forecasts for 5
to 6 weeks. Knowledge of how the PFM affects the loop current is important to the circulation of
the GoM both as a direct and indirect generator of surface-layer eddies and as a source of lower-
layer flows (Hamilton *et al.,* 2016). Loop current extension and anticyclonic eddy separation are
the result of the momentum imbalance (Pichevin & Nof, 1997) and form the shape of future LCEs.
Interacting seasonal and stochastic processes could trigger the separation of the LCEs (Fratantoni
*et al.*, 1998; Zavala-Hidalgo *et al*., 2003; Zavala-Hidalgo *et al*., 2006) as well as forming Caribbean
eddies and other topographic features (Garcia-Jove *et al*., 2016).  In this context, the LC system
has some similarities with the North Brazil Current retroflection (Pichevin *et al*., 1999; Zharkov
& Nof, 2010; Goni and Johns, 2001), the Agulhas retroflection (Baker-Yeboah *et al.,* 2010; de
Ruijter *et al.,* 1999) and with the Gulf Stream, where large meanders pinch off as warm rings
(Brown *et al*., 1983; Richardson, 1983; Savidge & Bane, 1999).
Despite extensive research, after more than a half-century we are still struggling to completely
understand LC variability, the processes controlling the loop current extension, and the mechanism
of detachment of anticyclones from the loop, which move CW.  Because positive time trends have
been reported in temperature, winds, sea level and the greater number of detached eddies separated



from the loop current, it can be expected that these phenomena would affect primary productivity
and, indirectly, surface chlorophyll concentration (Polovina, *et al*., 2008; Laffoley & Baxter.,
2016). In this work we reexamine the effect of the intruding CW in particular to better understand
how it affects primary production, and the surface spatio-temporal climatology.

**2. Data and Methods**
Three independent data sets were used to provide evidence of temporal variability in CW extension
in the GoM. We used absolute dynamic topography (ADT) and surface velocity fields (geostrophy
and Ekman) from the GEKCO (Geostrophic Ekman Current Observatory, Sudre *et al.,* 2013)
product with a resolution of 0.25˚x0.25˚, in conjunction with *Chl-a* ocean color data derived from
the reprocessing R2014.0 product suite from Aqua MODIS (Moderate Resolution Imaging
Spectroradiometer) and from SeaWIFS (Sea-Viewing Wide Field of view Sensor), using the OCx
Algorithm with a spatial resolution of 9X9 km (https://oceancolor.gsfc.nasa.gov/cgi/l3).
Climatology was created from maps of absolute dynamical topography (ADT) that result from the
elevation of the sea surface height referenced to the geoid using the product from the Data
Unification and Altimeter Combination System available on the AVISO (Archiving, Validation
and Interpretation of Satellite Oceanographic data) website
https://www.aviso.altimetry.fr/en/data. The ADT climatology was constructed using the 25 years
of daily satellite maps, averaging all the Januaries, Februaries … and Decembers. In this work, we
considered eddies in any stage of formation, detaching and reattaching to the Loop Current as an
evidence of the incursion of the CW. After the ADT climatology was obtained, the predominant
boundary contour of CW was extracted from each climatological month. It was observed that the
40 cm ADT was well matched to the climatological maxima of its respective EKE. For this reason,



the ADT 40 cm contour is taken as the main ADT reference that tracks the Caribbean Water Front
(CWF) contour.
Specifically, monthly CWF positions were obtained from short-term running averages of daily
satellite observations in three-year periods. Each running average was moved rearward by one
year, e.g. 1993-1995, 1994-1996 … 2014-2016, 2015-2017. For each three-year period, a set of
12 monthly maps was obtained resulting in a total of 23 sets of monthly CWF maps: 10 sets from
1993 to 2002 and 13 sets from 2003 to 2017.  We used the 40 cm contour of each set of three-year
averages because this was the contour with the highest EKE observed in the 25 year data set. To
retrieve the CWF contours, we first determined the initial latitudinal position of the CWF to be at
80.7 ˚W with the respective corresponding longitudinal positions between Cuba and Florida. The
CWF contour lines that run from east to west and finish close to the tip of the Yucatan peninsula
were separated by 0.2 ±0.1 degrees. However, some ADT contour "islands" appeared next to the
CWF with a typical distance of >0.3 degrees from the CWF contour. Additionally, a spectral
analysis was done using a daily time series of 25 years of ADT data to build a spatially averaged
region influenced by the LC between 91.25ºW, 23.125ºN and 83.5ºW, 28.12ºN.
When ADT Island distances were > 0.3 degrees from the front, we used a Matlab code procedure
to eliminate them from the CWF contours. Once the CWF's contours were retrieved, the next step
was to visually corroborate the quality and coherence of each CWF contour over the monthly field
maps of ADT, sea surface currents, and *Chl-a* distribution.   In this way, inconsistencies were
detected and corrected.   The Matlab code procedure satisfactorily corrected 91.3% of the
contours.  The remaining sets were corrected by hand via visual analysis.





• The wind stress product for the 01/01/1993 – 27/10/1999 period was obtained from (1)
https://www.ncdc.noaa.gov/data-access/marineocean-data/blended-global/blended-sea-
winds, and (2) for the 28/10/1998 – 20/03/2007 period from http://cersat.ifremer.fr (MWF
L3 daily QuikSCAT product) and (3) for the 21/03/2007 – 31/12/2017 period from
http://cersat.ifremer.fr/data/products/catalogue (MWF L3 daily ASCAT product),
• Surface ocean currents, geostrophic and Ekman currents, were taken from 25 years of daily
satellite maps from the 1993 - 2017 GEKCO products.
• The 2003-2017 monthly *Chl-a* ocean color product was derived from Aqua MODIS and
the 1998-2002 monthly *Chl-a* ocean color product was derived from SeaWIFS.
• Main mesoscale instabilities were obtained from calculations of the climatological monthly
EKE maps of geostrophic and Ekman currents obtained from 25 years of daily satellite
observations of GEKCO using following equation:
$$u = u' + U; \qquad u' = u - U$$
$$v = v' + V; \qquad v' = v - V$$
$$EKE = ½ (u'^2 + v'^2)$$

Where $(u, v)$ is the total current ($u = u_E + u_g$ and $v = v_E + v_g$; $(u_E, v_E)$, is the Ekman and $(u_g,$
$v_g)$ is the geostrophic current); $(U, V)$ are the means of the oceanic currents and $(u', v')$ are
the anomalies of the current. Ekman and geostrophic current components were obtained



from the GEKCO product (Sudre *et al.*, 2013). To look for a relationship between ADT
and EKE patterns, the 40 *cm* ADT isoline was overlaid on the monthly EKE maps to make
EKE means be representative of the energy of the mesoscale eddy field (Jouanno *et. al.*,

168     2012).

•  Assuming that the total current is the sum of the individual contributions of Ekman and
geostrophic currents, it is possible to calculate how much each component of the total
current contributes to the GoM circulation. The Absolute Ratio (*AR*) between the Ekman
and geostrophic fields for each climatological month is computed using the following
formulation:

$$AR = |u_E| *100 / \left(|u_E| + |u_g|\right) \tag{1}$$

where $|u_E|$ and $|u_g|$ are the magnitude of the Ekman and geostrophic field current
respectively. Values of *AR* close to 0 correspond to a total geostrophic dominion and 100
correspond to a total Ekman dominion. Applying the above formula (1), monthly maps of
*AR* were made using the Ekman and geostrophic field satellite observations (GEKCO).
For consistency between the different satellite datasets, all monthly climatological spatial fields
were standardized at 0.25˚x0.25˚ spatial resolution by bilinear interpolation.

**3. Results and discussion**
**3.1. Tracking the Intrusion of Caribbean Water**
The CW enters the gulf through the Yucatan Channel and exits through the Straits of Florida,
penetrating northward into the GoM until instabilities form in the current and a ring-like LCE



pinches off. There are two ways of tracking the CW: 1) tracking the thermal signal (not possible
in summer due to weak thermal contrast in the GoM), and 2) tracking the sea surface height trough
the satellite altimetry. In 2005, Leben, using the 17 *cm* contour in the daily sea surface topography
maps (this contour closely follows the edge of the high-velocity core of the LCEs and LC), tracked
the LC thermal fronts in the sea surface temperature images during good thermal contrast. In a
different way, Lindo-Atichati *et al*., (2013) calculated the maximum horizontal gradient of the sea
surface height (SSH) to track only the contours of the LCF. In this work, we used the ADT to
track both the LC and the LCE´s from the contours formed by the influence of the CW. Monthly
mean surface oceanic currents from GEKCO overplotted on the ADT data are shown in Fig. 1.
Maximum satellite surface current velocities in the Caribbean Sea and the GoM, as well as in the
Yucatan current on the continental coast, were >50 *cm s$^{-1}$*, coinciding with *in situ* estimates of ~
60 *cm s$^{-1}$* (Badan *et al*., 2005). The monthly GoM total current fields show the variability of the
primary forcing that coincides with the mean ADT edge; the vectors of maximum velocity are
tangent to the edge of the maximum slope change. To locate the CW, the 40 *cm* mean ADT's
isoline was chosen. The ADT reference corresponds to regions of maximum gradients of ADT,
and maximum EKE (*vide infra*). Fig. 1 shows that (mostly) in autumn and winter, the CW retracts
to its most southeasterly location. In contrast, in spring and summer, CW penetration moves
towards the northwest. In fact, the extension begins in May and reaches maximum penetration in
September, showing an annual pattern.
It is accepted that the LCE's occur in a geographical control zone, which is based on momentum
imbalance (Pichevin & Nof, 1997; Nof, 2005), rather than instability. Also, we should not abandon
the idea that the formation of instabilities such as meanders and cyclonic eddies are due to high
EKE produced by upstream conditions that influence the circulation within the GoM (Oey *et al*.,



2003) and produce changes in the fluxes in the Yucatan Channel (Candela *et al.*, 2002), transport
variations in the LC (Maul & Vukovich, 1993), variations in the deep outflow (Bunge *et al.*, 2002),
and cyclonic eddies in Campeche Bank and Tortugas (Fratantoni *et al.*, 1998; Zavala-Hidalgo *et*
*al.,* 2003). The areas of large EKE are related to the intrusion and retreat of CW and LC frontal
eddies (Garcia-Jove *et al.*, 2016) via baroclinic and barotropic instabilities (e.g. Jouanno *et al.,*

213    2009).

Fig. 2 shows that the 40 *cm* isoline encloses the maximum EKE area of the LC-LCEs during each
climatological month, demonstrating that its distribution is mainly centered in the LC region;
consequently, the maximum EKE borders the Caribbean water front just where the abrupt
horizontal gradients of ADT exist and changes of current speed occur. It is clear that the 40 *cm*
isoline of ADT matches very well both the maximum EKE values and the maximum ADT gradient
and is a good tracker of the contours of LC-LCEs. Lindo-Atichati *et al.*, (2013) proposed a
methodology using the SSH maximum horizontal gradient, which is the addition of sea height
anomaly and mean dynamic topography, to obtain the contours of LCF and the LCE's. In our
analysis, we chose the 40 *cm* isoline as a general reference to track both LCF and LCE´s
transporting CW. The enhanced monthly EKE signals respond in the same way as the LCF,
repeating the mean monthly pattern as well as the total currents; the CW intrusion starts in spring
and peaks in summer to retract in autumn and winter, and there are no relevant mesoscale EKE's
structures in the western GoM. These results confirm an annual pattern of CW intrusion in summer
months and retraction in winter.
**3.2 West and Northward Caribbean water extension**
The monthly intrusion of the CW was tracked by taking as a reference the Northward and





Westward positions of the 40 *cm* isoline of the Caribbean Water Front (hereafter CWF; Fig. 3).
During the winter months (January to March), the north position of the CWF (Fig. 3a) was
approximately 26.5 °N, reaching a maximum in August, to 28°N, and then decreasing in December
to 26.28 °N. The range in km between the lowest and highest north position of the CWF was 191
*km* or 1.72°. In regards to the Westward positions (Fig. 3b), the CWF during January, February,
March and April was at 88.19, 88.33, 88.23, 88.18°W respectively.  In May, the CWF quickly
stretched and in July, August, and September reached 90.2, 90.26 and 90.13°W respectively to
peak in October at 90.76 °W. The westward intrusion from the minimum to the maximum positions
was 2.57°, equivalent to 254 *km* (calculated at 27.5°N latitude). These results confirm the intrusion
of the CW for an annual period as follows: 1) Analysis of the maximum north and westward
penetration of the front over 25 years shows that from January to February, it is retracted southeast,
to ~ 26.55°N and ~ 88.32°W (Fig. 3a and 3b, respectively), 2) an ADT spectral analysis carried
out in the CWF region shows a strong annual signal that describe the back and forth of the ADT
signal during 25 years of daily data (Fig. 3c).
During April to June, the front advances more slowly in the northwest, progressively reaching
latitudes of 26.78°N, 27.09°N, to 27.27°N, with the respective longitudinal positions 88.2°W,
88.5°W and 88.99°W. In summer, the CWF intrudes the furthest into the interior of the GoM. The
front lengthens and slightly bends towards the west; its maximum southern and westward advance
occurs in August, reaching up to 28°N and 90.45°W. In October, the front penetrates west to
90.76°W, but decreases in latitude to 27.28°N. Finally, in December, the CWF retracts to its
southerly position near 26.28°N and 88.43°W.
From March to July, the CWF shows a north incursion to the north of 1.38°, equivalent to 153.6
*km*, with a penetration speed on the order of ~ 1.02 *km day$^{-1}$*. On the other hand, the rate of



retraction from August to October is ~ 1.86 $km\ day^{-1}$. Otherwise, the west penetration of the front
was 254 $km$ (Fig. 3b). The entire process of north intrusion occurred in three stages: first, the front
travels 205 $km$ into the GoM with a velocity of 2.3 $km\ day^{-1}$; then between July and August the
front remains quasi-stationary between 90.45˚W; finally, in September, it goes from 90.13˚W to
90.76˚W, equivalent to 63.6 $km$ at a rate of 2.1 $km\ day^{-1}$. The west retraction happens relatively
quickly as the front retracts 192.7 $km$ towards the east in a single month (October) at the rate of
6.4 $km\ day^{-1}$, while in November it travels 48.8 $km$ at a rate of 1.4 $km\ day^{-1}$. The effect of the
inclusion of eddies in the statistics shows their fingerprint in large western GoM areas as
anticyclonic circulation (Fig. 3b). This effect is more marked since 2002 when the number of
eddies per year began to increase. This will be discussed later.
Summarizing from  25 years of daily satellite observations, we note that the CWF floods annually
into the GoM during the warmer months (May to October) and retracts in the colder months, from
December to March. This annual periodicity is confirmed by the spectral analysis of the ADT
signal in the CWF region, which shows a highlighted frequency of 1 cycle per year. In addition, at
and near the surface, the CW   is of lower density than the common water of the gulf (GCW); and
its concentration of nutrients in the first 700 m is lower than the gulf water at the same depths, as
indicated by   comparing the internal region of the GoM and the Caribbean Sea using CARS2009
climatologies, (analysis not shown in this work).
In Fig. 4, the climatological ratios between the areas of standard deviation (STD) of the CWF
contours > 15 cm (dotted line) and CWF contours > 40 cm (heavy black line) were computed.
Ratio values greater than 1 were found in February (1.62) and April (1.60). On the other hand,
from May to August, the ratios were from 1.41, descended to 1.36, and increased in September to
1.60, peaking in October to 1.68, then decreasing in November to 1.60 and finally to 1.62 in



December. Winter had a high average ratio of 1.56.  In the last two months of spring and the first
two months of summer the average ratio was 1.40. However, in the last summer month the ratio
increased abruptly to 1.60. Fall kept high ratios, averaging 1.62. The complete time series of
monthly ratios shows an annual cycle having a valley between spring and summer and cresting
between fall and winter.

**3.4 Monthly Spatial Variability of the CWF and Wind Effect**

It was found that where penetration-retraction of the CWF occurs, variability varies from 15 to 35
*cm*, extending west to 90.8°W in winter and 93.5°W in summer (Fig. 4). West of the CWF, in the
deep zone of the GoM, the observed variability was close to 10 *cm* distributed in the band of
latitude between 23˚N and 28.5˚N. The regions of maximum variability (STD > 15 *cm*) occur in
the CWF zone and extend outside the irregular area of reference (isoline of the 40 *cm* ADT). The
effect of CWF penetration and regions of anticyclonic circulation was determined from the area of
the variability of ADT, with maximum values close to ~35 *cm* in the central region of the CWF,
at 86.67˚W and 25.6˚N. The percentage of the area of influence of STD > 15 *cm* in relation to the
area of the gulf ($1.56 \times 10^6$ *km²*) is presented in Fig. 5, where a gradual monthly increase is observed
from January to October, followed by a decrease in November and December. In January the direct
influence of the CWF on the gulf by area was 12.4%, rising to 21.5% for October, to subsequently
decrease in December to 15.4%.  The greater percentage area of the STD of the ADT may be
attributed to a  greater  influence of Caribbean Sea waters.
Regarding the effect of wind stress, the eastern component of the NE Trades blows throughout the
year, both in the gulf and the north Caribbean Sea.   However, the north component of the wind





stress presents notable variability, having a northerly component with a force of ~ 0.033 $Nm^{-2}$
during autumn-winter and reversing during spring-summer (April to June) to a constant intensity
of ~ 0.016 $Nm^{-2}$ (Fig. 6). During winter, the greater intensity of the northerly wind, brings deep,
nutrient-rich, water   to the photic zone and enhances the superficial *Chl-a* concentration.
The variability of the ADT signal indicates a displacement of subsurface water, mostly laterally,
while the larger spatial coverage of STD in late summer and early autumn indicates a facilitated
release of LCE's. In addition, the seasonal pattern of intrusion and retraction of CWF is related to
the high (15 <STD <35 *cm)* variability observed in the ADT maps of the LC-LCEs region. As the
CWF moves toward shore in the regions of the continental shelf, variability reduces to <10 *cm*,
possibly because mesoscale processes are attenuated in the continental shelf and other dynamics
may be present (Martínez-López & Zavala-Hidalgo, 2009).
**3.5 Geostrophy vs A-Geostrophy Balance**
Large Ekman (ageostrophic) areas stand out in the bay of Campeche and its continental shelf in
late summer, autumn, winter, and early spring, covering large portions of the total GoM area
(Fig.7). In contrast, the LC presents a markedly strong geostrophic dominance with an Absolute
Ratio (AR) close to 0 during each climatological month. In the western GoM, in the Bay of
Campeche, a positive cyclone emerges in September as a weak geostrophic structure and becomes
stronger into the autumn months to form a larger cyclone. Correspondingly, the CWF path is also
clearly seen in the AR fields that reaches its peak in summer and retracts in autumn. Twenty-five
years of continuous data of Ekman and geostrophic currents GEKCO make the above observations
robust.
**3.6 Changes in the Caribbean Water Incursion from 2003 to the Present**





Using the 40 *cm* reference, a 3-year running average of the ADT data was calculated to extract the
minimum number of years that would produce a similar pattern over a quarter century of the CWF.
The results have shown a difference in CWF path and westward penetration before and after 2002.
It is observed that before 2002 the CWF was less intrusive in the west of the GoM (Fig. 8),
however, after 2002 the CWF's extended to the west (Fig. 9). It is important to note that the
intrusion of the CWF is due to the influence of the LCE's that have a strong presence in the western
GoM. This fact is supported by a statistical analysis of the life of the LCEs in two time periods
(1993-2002 and 2003-2015) (http://www.horizonmarine.com/loop-current-eddies.html). The data
shows that the LCE's in the 1993-2002 period had a mean life of 6.8 months while the average life
in 2003-2015 was 11.7 months.  To prove that there is significant difference between these periods,
a student-*t* test was applied with the result that the difference between them is significant (t = -
3.098, p = 0.005). The LCE mean life difference is clear evidence that the incoming volume of
water from Caribbean Sea (with oligotrophic features, Aguirre –Gómez & Salmerón-García, 2015)
has reached farther in the western GoM after 2002. These observations also agree with the results
of Lindo-Atichati *et al.* (2013) confirming that, on average, the LC northward intrusion starts to
increase in 2002. This suggests that from 2003 onward, larger volumes of oligotrophic waters from
Caribbean Sea have invaded the western GoM.
**3.7 AMOC**
Our analysis suggests that extensions of the LC-LCEs in the GoM are related to an increase in the
volume of Caribbean Sea water that enters the GoM. Caesar *et al*., (2018) highlight that
"weakening of the AMOC by about 3±1 Sv caused by the warming effect consisting of a pattern
of cooling in the subpolar Atlantic Ocean and warming in the Gulf Stream region can be explained
by a slowdown in the Atlantic Meridional Overturning Circulation (AMOC) and reduced




northward heat transport, as well as an associated northward shift of the Gulf Stream" (Caesar *et*
*al*., 2018; Robson *et al*., 2014). Besides the warming of surface water, the decrease in AMOC
influences the frequency and intensity of hurricanes (Yan *et al*., 2017) and causes southern shifts
in the tropical rainfall.  On a synoptic scale, weakening of the AMOC reduces the poleward
transport of heat in the North Atlantic and triggers the phases of the Atlantic Multidecadal
Oscillations (AMO) that have been correlated to surface warming and positive Sea Height
Anomaly Residuals (SHARs).  Supporting our hypothesis, an evidence obtained from a data
analysis of daily transports at the Florida cable shows a 3 Sv decline in outflow and an increase in
sea level in the Florida Strait from 2004 to 2014 (Park & Sweet, 2015).

**3.7 *Chl-a* Satellite Imagery, Climatology, and Changes in the Last Decade**
Another product that tracks the effect of CW inside the western GoM is the *Chl-a* satellite imagery,
being an index of primary productivity (Boyer *et al*., 2009). Physical processes that affect the
distribution and abundance of *Chl-a* include estuarine influxes, depth of the nutricline, wind stress,
thermal stratification and eddy advection. However, over deep waters of the GoM, it is the wind
stress and the thermal stratification that principally affect the *Chl-a* concentration (Martínez-López
& Zavala-Hidalgo, 2009; Müller-Karger *et al*., 2015, Damien *et al*., 2018). It was found that the
oligotrophic Caribbean Sea waters contrast seasonally with the gulf waters and allow the
observation of two levels of *Chl-a*  (high and low, Müller-Karger *et al*., 1989). The temporal
relationship between the CWF and *Chl-a* concentration was constructed from SeaWifs and
MODIS monthly climatological images (Fig. 10). The highest concentrations of *Chl-a* in the
interior of the GoM are observed during autumn and winter months when high concentrations are
triggered by vertical mixing (Pasqueron de Fommervault *et al*., 2017; Damien *et al*., 2018) when





values were > 0.14 *mg m⁻³* in agreement with Dandonneau *et al.* (2004), whereas in spring-summer
they decreased to 0.08 - 0.09 *mg m⁻³*. During spring-summer, when the maximum CW penetration
occurs, our data confirms that the footprint of the CWF water (delineated by the 40 *cm* isoline of
ADT) is oligotrophic indicating that Caribbean water has indeed entered the GoM. During this
period, *Chl-a* surface concentration remains low as the increase in surface temperature strengthens
stratification. Additionally, the winds from the southeast are weak, thereby reducing the mixing
of nutrients to the surface. In contrast, during the autumn-winter months, the northerly winds are
stronger, increasing vertical mixing, deepening the mixed layer, and carrying cold, nutrient-rich
subsurface water into the euphotic layer (Müller-Karger *et al.*, 1991; Müller-Karger *et al.*, 2015;
Pasqueron de Fommervault *et al.,* 2017).
In seeking relationships between the spatial-seasonal distribution of the *Chl-a* concentration and
the incursion signaled by the ADT-generated data, three spatial-temporal periods were selected,
each was averaged pixel by pixel, and the three were labeled: "early" (1998-2002), "middle"
(2003-2008), and "contemporary" (2009-2014) epochs. The 5-year averages of the "early" and
"contemporary' periods of two separate areas were compared: 1) an area   located in the western
GoM at 95.5˚W, 22.12˚N and 91.5˚W, 25.87˚N, and 2) a smaller area located in the center of the LC
at 86˚W, 22.12˚N and 84.75˚W, 23.37˚N (Fig. 11). The differences in the means were tested for
significance with a 2-tailed z test at the 95% confidence level (Fowler *et al.*, 2013). The results are
shown in Table 1.

The results   in Table 1 may be summarized as follows:
**A**. Temporal differences: 1) Western GoM differences between "Early" and "Contemporary" *Chl-*



*a* concentrations are significantly different in all seasons; 2) Loop Current differences between
"Early" and "Contemporary" *Chl-a* concentrations are significantly different during winter, spring,
and autumn, but not in summer;
**B**. Spatial differences: 1) In winter, the Western GoM is significantly higher in *Chl-a* than the LC
during both "early " and "contemporary" periods; 2) In the spring, the Western GoM is
significantly higher than the LC during the "early" period, but not in "contemporary" period; 3)
In summer, the LC is significantly higher than Western GoM during both "early" and
"contemporary" periods; 4) In autumn, the Western GoM is significantly higher than LC during
the "early" period but not significantly different from the LC in the " contemporary" period.
**C**. Seasonal Differences. In the Western GoM and the LC in both the early and contemporary
periods, *Chl-a* decreases from winter to spring and from spring to summer, and increases from
autumn to winter, but autumn concentrations do not exceed winter. All differences are significant.

Examination of Table 1 indicates that at both areas, the winter season is most productive, followed
by autumn, with the lowest *Chl-a* concentrations occurring in summer. There is also a time-
dependent trend, with contemporary values that are, in general, lower than the values in the early
and middle epochs. Both areas exhibit identical climatic trends over time and during each season,
indicating that these effects are applicable outside of the continental shelf. The "early" spring
epoch is more eutrophic than the middle and contemporary epochs, indicating a decline in nutrient
concentrations over time. This is also evident in the LC core, where *Chl-a* concentrations also
decreased with time and signals the entrance to the gulf of more water oligotrophic during the
middle and contemporary epochs. Perhaps the most dramatic seasonal scenarios occur in summer

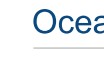 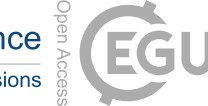
to early October period, when the CWF "tongue" invades the interior of the GoM and extends over
the deep waters. Although the concentration of *Chl-a* in the Western GoM declines gradually with
time to from ~ 0.09 to ~ 0.08 *mg m⁻³*, the interesting fact is that the area of oligotrophic water
expands and become larger in the contemporary period. On the other hand, the LC core *Chl-a*
concentrations in the three epochs do not significantly differ, suggesting that the water entering
the GoM is from a single source, namely, the Caribbean Sea. In general, the extensive penetration
of the LC within the GoM, as well as the increase in the life periods and sizes of eddies coincide
with the intrusion of nutrient-poor Caribbean Sea water.
Comparatively and as mentioned, autumn and winter have the highest *Chl-a* concentration due to
enhanced vertical mixing of colder waters due (partially) to the intensity of the wind blowing from
the north. Although the differences are not great, the area of low concentration is considerably
larger during the middle and contemporary periods than the early epoch (Table 1, Fig. 11). Also
waters from the Caribbean Sea penetrate further in autumn into the interior of the GoM than they
do in winter. Two points summarize the result of the seasonal analysis of the three epochs: First,
the extent of the CW intrusion confirms the north-west eddy migration during each epoch, second,
the *Chl-a* concentration declines over time.
The second point was confirmed by calculating the average *Chl-a* concentrations outside the
continental shelf over two time periods, considering only the concentrations above waters deeper
than 250 *m*. Using data from 1998 to 2002 (SeaWIFS), and from 2009 to 2014 (MODIS) we
conducted a student-t test for difference in the means (Fig. 12). The latter period was significantly
lower with t=4.75 and p<0.001 (n₁ = 1,825; n₂ = 2,190). This analysis confirms that the *Chl-a*
concentration of the GoM decreases over time and appears to disagree with the results of Müller-
Karger *et al.* (2015) who did not indicate a time trend in *Chl-a* concentration in the GoM  As  the



data were taken with different sensors   and to eliminate the uncertainty that this difference is not
caused by a systematic difference between the SeaWIFS and MODIS data sets used, we calculated
least square regressions to the SeaWIFS and MODIS time series (Fig. 13) at four stations
corresponding to the northwest, northeast, southwest and southeast regions of Müller-Karger *et al*.
(2015) (hereafter M-K)). For each data set, inner slopes as well as overall slopes were calculated.
For all four stations, the SeaWIFS (1998-2002) and the MODIS (2003-2017) data series merged
exactly and all stations show negative trends; equivalently, the combined time series (1998-2017)
also show a negative tendency, supporting the conclusion that the *Chl-a* concentrations over the
deep GoM has decreased over time.
The difference between our results and those obtained by M-K may be attributed to the different
way in which we and M-K treated the data. M-K divided the GoM into 4 quadrants with depths of
over 1000 m: Region 1-North East (RO1), Region 2 (RO2 -Northwest), Region 3 (RO3-Southeast),
and Region 4 (RO4 Southwest) and calculated the spatial average in each quadrant to build four
time series, from 1993 to 2012. In their words, "Time series of anomalies of wind speed, SST,
SSHA and *Chl-a* concentration were obtained by subtracting the monthly mean (climatology) from
the monthly field for that variable".  Time series of wind "intensity", sea surface temperature
(SST), sea-surface height (SSH), and *Chl-a* data obtained at these stations from satellite products
was analyzed statistically, and plotted. Other variables plotted by M-K were mixed layer depth
(MLD) as calculated from a hydrodynamic model, and net primary production (NPP) calculated
from MODIS data using the vertically generalized production model (VGPM) of Behrenfeld &
Falkowski (1997).
On the other hand, we calculated the average of the *Chl-a* concentration pixel by pixel in waters
over 250 m depth, for two time periods (1998-2002 and 2008-2016), and subtracted the respective





monthly (climatological) means to find the difference (Fig. 12). From 2010 onward, the difference
indicated a small reduction of *Chl-a* in the first optical depth (1-20 or 40 meters of depth) that is
increasing with time. A student-t test was used to conclude that the reduction was significant.  We
also treated the data exactly as M-K did and obtained slightly negative slopes over the entire 1998
to 2013 period.
We suggest that M-K did not detect the small negative trend in their *Chl-a* plots because their
calculated  slopes  indicated no time-dependent change. We surmise that they were also influenced
by the lack of slope in the modeled MLD plot, despite clear, positive trends for SST, SSHA, and
wind force.  Actually, although close to zero, the slopes, as indicated in M-K, were not zero, but -
0.03 for RO1, -0.01 for RO2, and simply given in as -0.0 for RO3 and 0.0 for RO4 (see Table 1 in
M-K). They also ignored the fact that the time-*Chl-a* correlation coefficients (R) for all four regions
were negative.
To confirm our findings, we chose 4 stations, each one centrally located in each M-K quadrant,
and conducted regression analyses of the logarithmic transform of the SeaWifs and MODIS *Chl-*
*a* concentrations. All four regions showed a negative slope, a negative R, and the negative slopes
in the southern gulf (RO3 and RO4) were significantly different from 0 ($p \ll 0.05$). This is shown
in Fig. 13.
The observed small, but persistent decline in *Chl-a* from 1993 to 2017 may be attributed to the
AMOC's over-all effect of warming the surface water and thereby promoting stratification.
However, we wish to make clear that our conclusion about the recent time-dependent lowering of
the *Chl-a*   pertains only to the near surface, and may not indicate a decrease in integrated water
column primary productivity. In the GoM, the chlorophyll maximum as measured by fluorescence
occurs at about 75 m, e.g., below one optical depth, and is greater in summer than in winter



(Pasqueron de Fommervault *et al.,* 2017), indicating that the relationship between water column
productivity and near surface *Chl-a* concentration in the GoM begs further study.

**4. Summary and conclusions**
In this work, the inference of CW intrusion was evaluated using a large data set of satellite-derived
observations. The availability of a large spatial extension of satellite observations of ADT, sea
surface currents, wind stress over a quarter of century, and *Chl-a* over 20 years has enabled us to
confirm the temporal pattern observed in the 60's and 70's with more recent *in situ* observations.
The verification of the CWF climatologies developed in this work is important as a reference
baseline for further numerical modeling, and it impacts assessments of the gulf's biogeochemistry,
energy, heat transport, and *Chl-a* concentration.
As a point of interest, a recent committee  (National Academic of Sciences, 2018) suggested three
main study topics to advance the knowledge of the processes that characterize the GoM: 1) the LC
system active area, 2) the variation of the inflows of the LC system, and 3) the dynamic interactions
of the LC system in the west.  In this study, we examined all three issues using a quarter century
of remotely sensed satellite data. Based on these, we have confirmed that the maximum influence
of the CW into the GoM (e,g., its maximum extension into the gulf or intrusion) has a temporal
variability, being stronger in summer and weaker in the late fall and winter. This is supported by
the fact that the generated monthly EKE maps have the maximum gradient at the periphery of the
CWF and have a similar monthly pattern of extension and retraction as the CWF
We noted that in the summer months the wind stress from the southeast is weak, thereby
minimizing the flow of nutrients to the surface and causing *Chl-a* to be  low , specifically for three
reasons: 1) The increase in the surface temperature of the water column strengthens  stratification



2) The intrusion of the CW to the western gulf´s surface, thickens the surface layer, and  3) The
eddy-driven anticyclonic circulation deepens the nutricline.  This contrasts with the cold seasons,
when the surface temperature of the water is lower and the northerly winds are stronger, favoring
the flow of nutrients to the surface.
The three-year running averages of ADT 40 *cm* isoline reproduce qualitatively the climatological
pattern of a quarter of a century showing that before 2002 the CWF was less intrusive and the LCE
sizes were smaller. In the 1993-2002 period, we calculated that the eddy mean lifespan was 6.8
months and that in the 2003-2015 period the mean lifespan was 11.7 months. This difference
suggests that after 2003, larger volumes of oligotrophic waters from Caribbean Sea have invaded
the western GoM and reduced mean surface *Chl-a* concentrations.
In summary this work shows that
•   The LC-LCEs influences and enters further into the western GoM than was previously
514          known.

•   Within the CWF, the ADT 40 *cm* isoline borders the *Chl-a* gradients.
•   *Chl-a* concentrations respond to the dynamics inside the GoM and are influenced by
517          the CWF and the LC anticyclonic and cyclonic eddies.

•   Since 2002, near surface *Chl-a* concentrations over bathymetry deeper than 250 m
519          have decreased, and GoM surface waters may be turning more oligotrophic than in the
520          previous decade.

This work, based on 25 years of remotely sensed data, emphasizes the role of climatology in

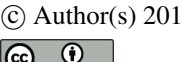



determining GoM circulation and its productivity and suggests that further climatologically-
induced changes are probably imminent.

**5. Acknowledgements**

This study was carried out as part of the PhD thesis research conducted by the lead author at the
Faculty of Marine Science and the Oceanographic Research Institute (FCM-IIO / UABC),
Postgraduate Coastal Oceanography Program, and it was supported by the Graduate Professional
Development Mexican Program grants (PRODEP: DSA/103.5/16/5801), the National Institute of
Technology of Mexico (TecNM) and the Mexican Energy Bureau and Hydrocarbons Mexican
Trust, project 201441. This is a contribution of the Gulf of Mexico Research Consortium (CIGoM).
Altimeter products were produced by Data Unification and Altimeter Combination System
available on the AVISO (Archiving, Validation and Interpretation of Satellite Oceanographic data)
https://www.aviso.altimetry.fr/en/data. Chl-*a* maps were derived from Aqua MODIS (Moderate
Resolution Imaging Spectroradiometer), https://oceancolor.gsfc.nasa.gov/l3/ and SeaWIFS (Sea-
Viewing Wide Field of view Sensor), using OCx Algorithm with a spatial resolution of 9X9
https://oceancolor.gsfc.nasa.gov/l3/. Wind Stress, Geostrophic and Ekman Currents were
extracted from GEKCO (Geostrophic Ekman Current Observatory, Sudre et al., 2013)
http://www.legos.obs-mip.fr/members/sudre/gekco_form with support from LEGOS. In particular
for wind stress GEKCO product, they were used these three sources for 01/01/1993 - 27/10/1999
period https://www.ncdc.noaa.gov/data-access/marineocean-data/blended-global/blended-sea-
winds, for 28/10/1998 - 20/03/2007 period (MWF L3 daily QuikSCAT product)
http://cersat.ifremer.fr and for the 21/03/2007 - 31/12/2017 period (MWF L3 daily ASCAT
product) http://cersat.ifremer.fr/data/products/catalogue. Finally, the general features of the Gulf



of Mexico Loop Current eddies were taken from https://www.horizonmarine.com/loop-current-
eddies.



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



Table 1. Average *Chl-a* concentrations ($mg\ m^{-3}$) at two geographical areas: 95.5˚W, 22.12˚N and 91.5˚W,
25.87˚N, (Western GoM) and 86˚W, 22.12˚N and 84.75˚W, 23.37˚N (LC-LCEs) during **"early"** (1998-
2002), **"middle"** (2003-2008), and **"contemporary"** (2009-2014) epochs. Table 1 shows the compared
averages in bold print. Standard deviations and number of pixels considered are shown in parenthesis.

| Geographical Areas | Season | Early Averages (1998-2002) | Middle Averages (2003-2008) | Contemp. Averages (2009-2014) |
|---|---|---|---|---|
| Western GoM | Winter | **0.180** (±0.047, n=4026) | 0.167 (±0.048, n=4866) | **0.173** (±0.0624, n=4828) |
| Loop Current | | **0.149** (±0.052, n=536) | 0.129 (±0.064, n=647) | **0.117** (±0.062, n=645) |
| Western GoM | Spring | **0.114** (±0.033, n=3693) | 0.087 (±0.049, n=4658) | **0.0834** (±0.036, n=4754) |
| Loop Current | | **0.0948** (±0.074, n=526) | 0.085 (±0.1287, n=642) | **0.0835** (±0.116, n=648) |
| Western GoM | Summer | **0.0887**(±0.024, n=3924) | 0.080 (±0.022, n=4794) | **0.0755** (±0.023, n=4837) |
| Loop Current | | **0.109** (±0.217, n=535) | 0.091 (±0.171, n=647) | **0.0938** (±0.148, n=648) |
| Western GoM | Autumn | **0.151** (±0.052, n=3894) | 0.137 (±0.044, n=4876) | **0.127** (±0.043, n=4846) |
| Loop Current | | **0.138** (±0.128, n=525) | 0.1325 (±0.114, n=643) | **0.122** (±0.103, n=648) |












**FIGURE CAPTIONS:**

Fig. 1. Monthly means of absolute dynamic topography (ADT) and surface currents averaged over a quarter of a century (1993-2017).

Fig. 2. Climatological monthly maps of eddy kinetic energy (EKE) in GoM; red color contours correspond to the areas of maxima EKE. The thick black line corresponds to the isoline of 40 *cm* of the CWF. The EKE was calculated using daily maps of satellite-derived currents from AVISO (GEKCO) for a quarter of a century (1993 – 2017).

Fig. 3. Geographical positions of the CWF tracked using the 40 *cm* ADT isoline representing 1993-2017 monthly average values of the absolute dynamic topography (ADT): a) Northward and b) Westward, respectively; c) ADT spectral analysis in a region influenced by the CWF (91.25ºW, 23.125ºN and 83.5ºW, 28.12ºN).

Fig. 4. The ADT quarter-century CWF (1993-2017) monthly climatology and its standard deviation are shown in thick and dotted lines, respectively. The thick line corresponds to the 40 *cm* isoline of the CWF. The dotted line enclose values of the >15 *cm* standard deviation.

Fig. 5. Average monthly percentage surface areas CW in the interior of the Gulf of Mexico determined from climatology of the STD contour > 15 *cm*; enclosed areas were calculated in relation to the GoM area ($1.56 \times 10^6 \ km^2$).

Fig. 6. Meridional component of the long-term monthly means of the wind stress and monthly CWF (40 *cm* ADT): color contours represent the meridian wind stress intensity; CWF is represented by the 40 cm thick black line; black arrows represent the monthly wind stress field; This graph was made using daily fields of 25 years of AVISO satellite images, 1993 to 2017.

Fig. 7. Absolute Ratio (AR) of ocean Ekman (a-geostrophic currents) and geostrophic currents for each climatological month; green-blue colors correspond to geostrophic areas and yellow-red colors correspond to areas influenced by the a-geostrophic currents.

Fig. 8. Monthly means of absolute dynamic topography (ADT) from 1993-2002 (color) and its respective CWF computed with the 40 *cm* isoline (thick line).

Fig. 9. Monthly means of absolute dynamic topography (ADT) from 2003-2017 (color) and its respective CWF computed with the 40 *cm* isoline (thick line).

Fig. 10. Monthly climatologies of *Chl-a* (SeaWIFS, 1998-2002 and MODIS data source, 2003-2017). The thick line represents the isoline metric of the 40 *cm* contour that represents the CWF (1998-2017). *Chl-a* values larger than 1 *mg m⁻³* are plotted in red color.

Fig. 11. From top left to bottom right, the Chlorophyll-*a* mean periods: column 1, SeaWIFS 1998-2002, column 2, MODIS 2003-2008 and column 3, MODIS 2009-2014 are shown; from top to



bottom correspond to the mean seasons: winter, spring, summer and fall. Average *Chl-a*
concentration is computed inside the red and white squares (red rectangle correspond to the
western GoM area and the white one as LC area) for each epoch season and their values are placed
in Table 1.
Fig. 12. Differences of *Chl-a* concentration ($mg$ $m^{-3}$) between the mean periods 2009-2014 of
MODIS minus 1998-2002 of SeaWIFS   The black broken line represents the isobath of 250 $m$.
White contoured areas indicate no significant differences.
Fig. 13. *Chl-a* concentrations ($mg$ $m^{-3}$) at four stations (a to d) in the GoM, daily time series derived
from SeaWIFS from 1998 to 2002 (green Color) and MODIS from 2003 to 2017 (blue color).
Intrinsic least square regressions for SeaWIFS (red line), MODIS (cyan line), and the overall linear
regressions for each station (dashed black line).



















FIGURE 1






FIGURE 2



FIGURE 3





FIGURE 4



**ADT STD contour colors (dotted line 15 cm) & CWF (thick line 40 cm )**








FIGURE 5

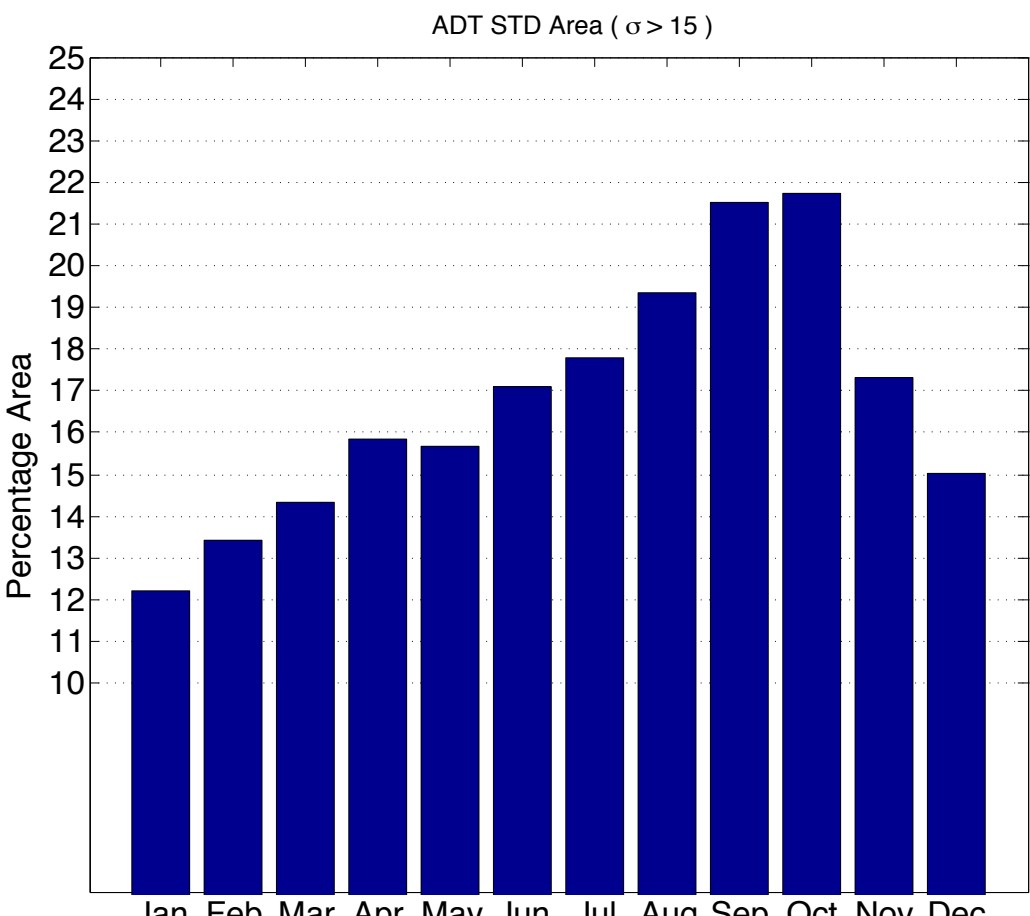











FIGURE 6





FIGURE  7




FIGURE 8






FIGURE 9







FIGURE 10





FIGURE 11




FIGURE 12

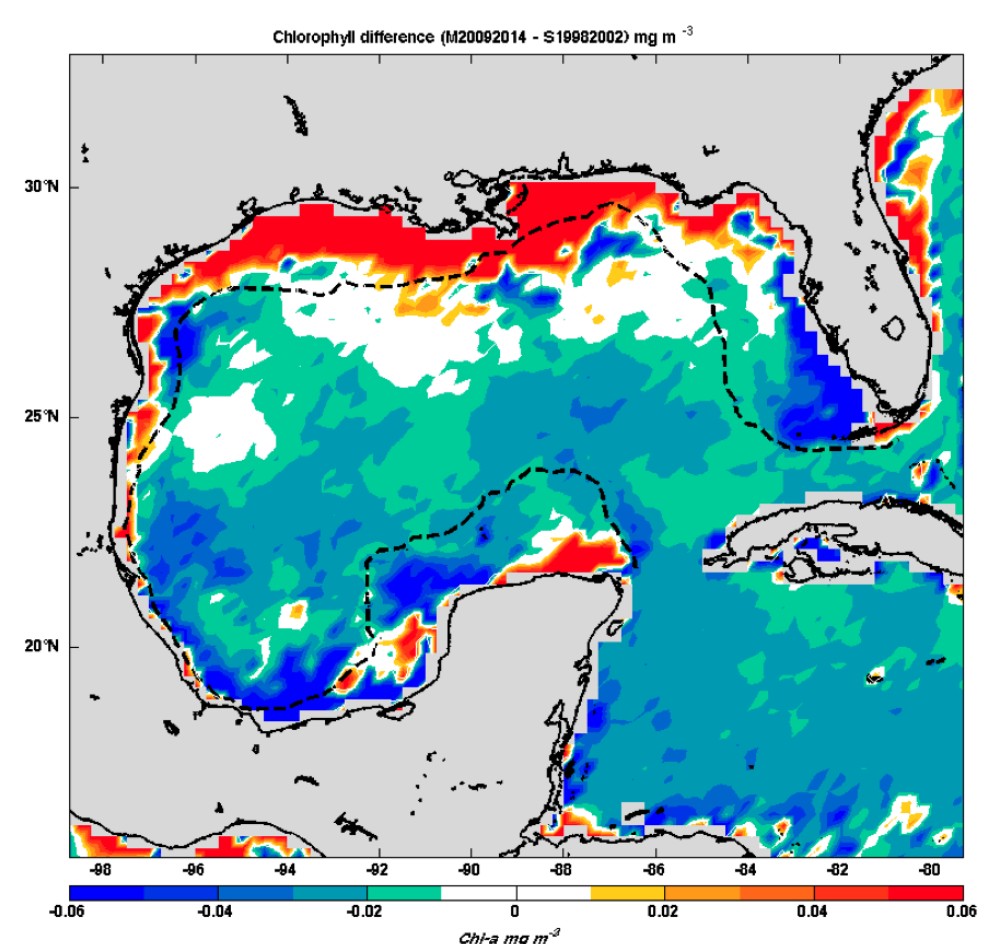












FIGURE 13

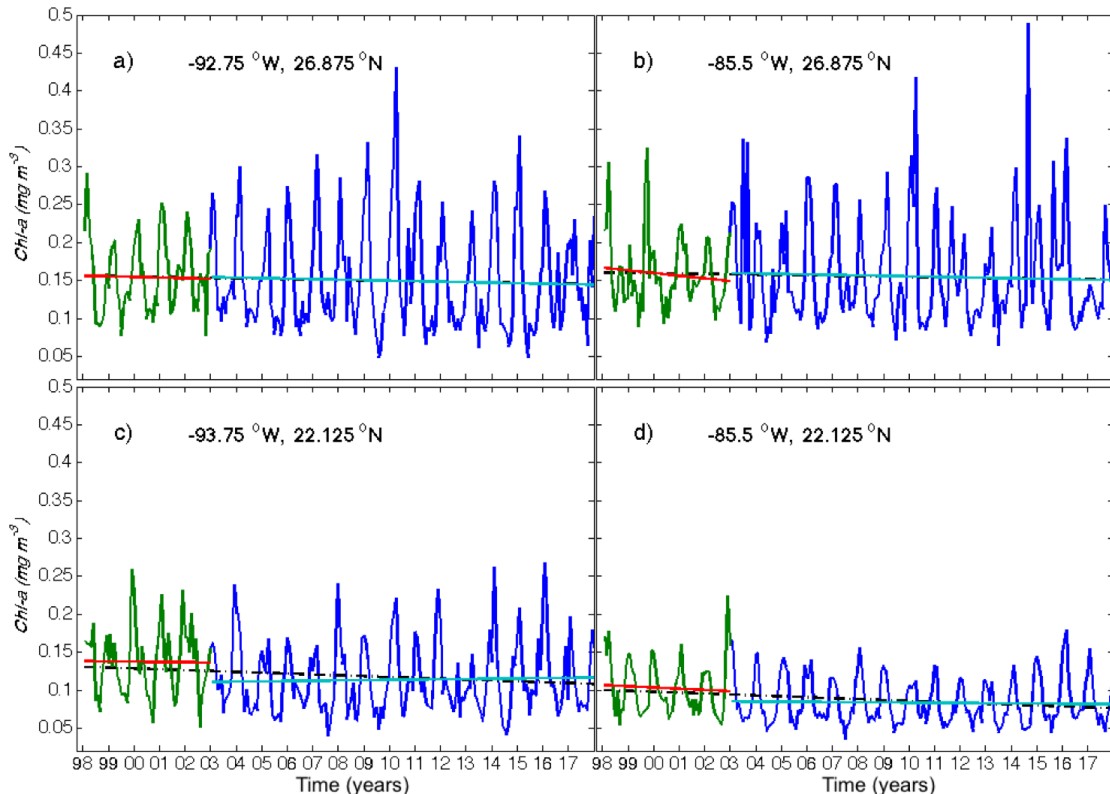

