# Peer review of "Effect of Caribbean Water Incursion into the Gulf of Mexico derived from Absolute Dynamic"

_Ocean Science, 2019_

## Referee Comment (RC1) · Anonymous Referee #1 · 13 Aug 2019

General Comments: The subject addressed is I think of great interest both within the regional oceanographic community and given the significant contribution of the LC to Gulf Stream transport of warm water to higher latitudes of general interest to the climate community. It could I think be improved (and refined) in some regards and some ill-supported or weakly supported speculations need to be more fully developed or eliminated. That said, the major conclusion (admittedly not a purely original one) that there is a predictable seasonality (and pattern) with respect to Caribbean water intrusion into the Gulf seems well grounded and it does appear that may have increased to

a small but measurable extent in recent years. I also like (and appreciate) the effort to include the contribution of LCE using EKE as a criteria to select a 40cm contour for the Carribean Water Front. Specific Comments: 1. A minor point is to what extent some aliasing may have been introduced by standardizing the satellite data into the same spatial resolution. 2. It is not clear that the authors are keeping in mind an inherent limitation of the data in that all the data sets they are analyzing are essentially surface or near surface (a small part of the overall circulation). This is germane to the comparisons made and between Eckman and geostrophic flow regime patterns as well as other issues raised. 3. While they properly conclude that their analysis "suggests" (see section 3.6) larger volumes from 2003 onward it is by no means conclusive (see comment above). 4. I have issues with section 3.7 AMOC both in that they proceed as if it were shown definitively that a greater volume of Caribbean water is entering the Gulf and their use of the Caesar paper. They also then elaborate upon AMOC and synoptic scales which is pure speculation and unrelated to their own analysis 5. With respect to the satellite chlorophyll data the authors do not appear to understand the limits of the data. It is not only that only surface (or near surface) pigment concentration is measurable by satellite, it is more fundamentally the case that changes in the measurement can be indicative of many things other than changes in plant biomass. There is particular sensitivity to changes in plankton community structure (therefore pigment type and concentration per unit biomass). Not only are some of the differences noted smaller than I for one would be comfortable as conclusive but in fact differences in community structure in many oceanic regions (including the GoM) have been widely reported and indeed are expected given warming, acidification and changes in nutrient loading. None of this is to say that over the deeper regions of the GoM plankton biomass has not decreased but it simply cannot be rigorously inferred from this analysis.

---

## Referee Comment (RC2) · Anonymous Referee #2 · 23 Aug 2019

General comments:

In this manuscript (MS), the authors evaluate different long lime series dataset (ADT, Chl-a, wind stress, geostrophic and Ekman currents) to describe the seasonal cycle and interannual variability of the Loop Current. They also relate this variability with the Chl-a. The MS enhance the existence of an annual seasonal cycle of the Caribbean Water (CW) intrusion, or Loop Current (CL) extension, which has already been discussed with altimetry data in previous studies. However, the seasonal averages of the Chl-a and the climatological pattern observed before and after 2002 is an interesting

finding that deserves to be considered for publication, after mayor modifications of the MS.

There is a lack of discussion regarding previous studies, which referrer to the seasonal cycle of the LC, like Chang & Oey (2012, 2013), which expose a bimodal cycle obtained with model outputs; Hall & Leben (2016), which show an annual cycle with altimetry data; or Candela et al. (2019), which estimate the amplitude of the seasonal cycles of the transport at Yucatan Channel and at the Strait of Florida using mooring observations. Those works, among others, must be at least mention in the introduction and their findings should be discussed with what was found in this MS.

The conclusion regarding the decrease of Chl-a concentrations in the western Gulf of Mexico (GoM), where the authors state that there is a reduction due to a larger volume of waters coming from the Caribbean Sea toward the GoM, should be taken with caution and considers/discuss the limitations of the data series used for this conclusion. There are other sources that could contribute to a reduction in Chl-a in the western GoM. It is not obvious to me what the authors state with the footprint of the CW/LC path, in the Chl-a climatological patterns (Fig. 11).

The MS need to focused to highlight the relevant contributions. Additionally, statistical relevance of the seasonal cycle needs to be deeply discussed; it is necessary to specify how representative are the climatological averages (Fig. 1), obtained from the monthly averages of the individual years (using information from Fig. 4). Description of section 3.2 (lines 229 – 262 and lines 271 – 280) needs to be rewrite in order to guide the reader to the months of the year that the authors are talking about (maybe a Table will be useful); What is the main message given by all these numbers and description? Besides, there need to be an agreement between the months of the year with the LC extension/retraction throughout the MS.

Specific comments

The MS focus the discussion on the oligotrophic CW intruding the GoM owing to the

Loop Current. Therefore the description of the water masses that conform the GoM, must be introduced and supported by additional bibliography, aside from Nowlin & McLellan (1967), since the reference of Schmitz (2005) is not referred to water masses composition. Further Tanahara (2004), cannot be consulted.

I do not understand why the Loop Current track is called Caribbean Water. Discernment must be used throughout the MS when the authors refer to the Caribbean Water and the Loop Current; CW could be used to name the water intrusion from the Caribbean with specific biological and physical characteristics (maybe when they talk about Chl-a characteristics) and the LC when you refer to dynamic characteristics, i.e. current that enters the Yucatan Channel, loops at the eastern GoM and through the Florida Strait.

Lines 66-67: I do not consider that the MS shows a detailed analysis of the Loop Current Eddies, maybe the approach could be focus to the analysis of the LC and the LCE path footprint.

Line 77: Who is this acting as a primary forcing mechanism of the Loop Current? Yucatan Current? The term PFM is not clear and must be specified.

Line 100: "which move CW"……..where?

Lines 104-105: "In this work we reexamine the effect of …." Rephrase please.

Line 121: "we considered eddies in any state of formation, detaching….", then they are Loop Current Eddies (LCE), not only eddies.

Lines 146-154: The first three points of the paragraph referring to data description are repetitive with the first paragraph of section 2; the description of the datasets is disordered, since they are described in two different places. I suggest to add additional information, such as the years of the data that you are using at the beginning of the section, before methods description. The calculation description of the mesoscale instabilities, as well as the AR can be described in the methods section, where they have already been described (without paragraph mark).

Line 164: Is repetitive with line 110.

Line 192: The contours from the ADT are not determined by the influence of the CW, please rephrase.

Please define which months of the year are described for each season (I guess winter (Jan, Feb, Mar), spring (Apr, May, Jun), etc.?). It is not clear to me, at least from Fig. 1, that the LC extension is retracted in autumn and extended in spring (lines 200-203), or the fact that the maximum penetration occurs in September, instead of August (see also your statement in line 232). I think it could be useful for the authors to discuss their results with previous work referring about the seasonality of the LC (see comment above).

Line 241 and Fig.3: Please specified if the ADT was spatially averaged. The steric signal included in the ADT data must be discussed, considering the high-energy observed in the annual period (see Hall & Leben, 2016).

Section 3.2 (Fig. 4): Why the STD contour of 15 cm was chosen as a reference for the regions of maximum variability? Lines 278-280: How this cycle of the monthly ratios compares with the results of Chang & Oey (2012, 2013)?

It makes not sense to me the discussion of the monthly averages of the wind stress, shown in Fig. 6 and described in section 3.4, it is not relevant for the main objective of this section, further this discussion do not reinforce de main idea exposed here; I suggest to delete this part or move this description elsewhere in the MS.

Lines 300-301: Please be more specific; Do you mean an upwelling? Is so where? Please use references.

The whole paragraph of section 3.5 is not linked with the rest of the MS, if you want to keep it, at least it go deeper in the implications and discussion of these calculations.

Statement of line 327-329: The difference between the mean life of the LCEs (6.8 vs 11.7) needs to be discussed with previous studies.

Lines 378-379: How these three periods were chosen? Why not using the same two periods of the ADT? Please explain.

Table 1 (lines 386-399), please specify what means the bold numbers in Table 1 and show the difference between Early and Contemporary periods at each row. Using the differences obtained here discuss the significance of the Chl-a averages between these two periods.

Second point mark of the conclusions (line 515). I do not see that in Fig. 10, or in Fig. 11.

Technical corrections:

Please avoid the double space throughout the manuscript (i.e., lines 58, 60, 144, 263, 267, 269, etc.).

Caribbean water is mentioned for the first time in the MS in line 68 as the acronym CW and in the line 75 as Caribbean Water. The acronyms must be defined for the first time as they are mentioned in the text and then they should be used throughout the MS as an acronym (the same for Loop Current as LC, which is even lowercase in line 99).

Line 119: Specifiy the years used for the 25 years climatologies.

Line 140: 'island' instead of Island.

Line 216 and 230: CWF instead of Caribbean water front, it is already defined in line 125.

Lines 294-295: Please specify that this is a supposition.

Line 314: Please use the accurate terms.

Line 324: 'extended to the west'. . .in summer and autumn?

Line 337: Avoid the use of acronyms in the title sections (especially if it has not been previously defined).

Line 360: Use CW.

Line 402: Add (see also Fig. 11).

Lines 413-415: Rephrase.

Line 432: Check the point marks.

Lines 446-450: Rephrase the way you mention the M-K work please.

Please avoid the use of 'we', instead use something like 'in this work. . .'

Line 470: R means correlation?

Line 480: Change 'begs' for a more appropriate word (needs, requires, etc.).

Lines 508 and 509: Please change the term 'lifespan'.

Line 513: Please rephrase the first point of the conclusions.

---

## Author Comment (AC2) · 21 Sep 2019

We would like to thanks the reviewer for the positive criticism and recommendations. Below you find a point-by-point response to all comments.

QR2-There is a lack of discussion regarding previous studies, which referrer to the seasonal cycle of the LC, like Chang & Oey (2012, 2013), which expose a bimodal cycle obtained with model outputs; Hall & Leben (2016), which show an annual cycle with altimetry data; or Candela et al. (2019), which estimate the amplitude of the

seasonal cycles of the transport at Yucatan Channel and at the Strait of Florida using mooring observations. Those works, among others, must be at least mention in the introduction and their findings should be discussed with what was found in this MS.

ANSWER: About the lack of discussion regarding previous studies which referer to the seasonal cycle we add in L98-105: "Loop current extension and anticyclonic eddy separation are the result of the momentum imbalance (Pichevin & Nof, 1997) and form the shape of future LCEs. Chang and Oey (2010) using a numerical model, proposed that the wind stress could be the primary forcing that releases LCE's. In a second paper, supported by satellite observations, they proposed that the LC intrusion and the shedding of the LCE's followed a biannual cycle (Chang and Oey, 2013). Recently, Candela et al (2019) analyzed four years of water current data and reported a seasonal cycle in the transport through the Yucatan channel with the annual cycle as the main harmonic peak in July and the semiannual peaks January and July".

QR2-The conclusion regarding the decrease of Chl-a concentrations in the western Gulf of Mexico (GoM), where the authors state that there is a reduction due to a larger volume of waters coming from the Caribbean Sea toward the GoM, should be taken with caution and considers/discuss the limitations of the data series used for this conclusion. There are other sources that could contribute to a reduction in Chl-a in the western GoM. It is not obvious to me what the authors state with the footprint of the CW/LC path, in the Chl-a climatological patterns (Fig. 11).

ANSWER: We agree with the reviewer and we add in L441 to L446. "Our result and conclusions are based on SeaWifs and Aquamodis chlorophyll data, which in Type One water, correlate very well with chlorophyll measured with standard laboratory methods (Mati Kahru, personal communication). In our work we can only say that according to these satellite "products", we find a time-dependent diminution of the chlorophyll signal. This diminution has been widely observed by others in other waters (Behrenfeld et al., 2006, Polovina et al., 2008; Irwin and Oliver, 2009, Laffoley & Baxter., 2016)" About the state with the footprint of the CW/LC path, in the Chl-a climatological patterns
(Fig. 11) we agree and we improve the line as follow: "During spring-summer, when the maximum CW penetration occurs, our data confirms that the footprint of the CWF water (delineated by the 40 cm isoline of ADT) is in general oligotrophic indicating that Caribbean water has indeed entered the GoM".

QR2-The MS need to focused to highlight the relevant contributions.

ANSWER: The MS highlight the relevant contributions thanks to all the recommendations.

QR2-Additionally, statistical relevance of the seasonal cycle needs to be deeply discussed; it is necessary to specify how representative are the climatological averages (Fig. 1), obtained from the monthly averages of the individual years (using information from Fig. 4).

ANSWER: We improve section 3.1 but also we rewrote section 3.2 which includes discussions about seasonality.

QR2-Description of section 3.2 (lines 229 – 262 and lines 271 – 280) needs to be rewrite in order to guide the reader to the months of the year that the authors are talking about (maybe a Table will be useful); What is the main message given by all these numbers and description? Besides, there need to be an agreement between the months of the year with the LC extension/retraction throughout the MS.

ANSWER: All section 3.2 was rewriten and throughout the MS we are careful to connect the months of the year with the LC extension/retraction.

Specific comments QR2-The MS focus the discussion on the oligotrophic CW intruding the GoM owing to the Loop Current. Therefore, the description of the water masses that conform the GoM, must be introduced and supported by additional bibliography, aside from Nowlin & McLellan (1967), since the reference of Schmitz (2005) is not referred to water masses composition. Further Tanahara (2004), cannot be consulted.

ANSWER: We introduce the description of the water masses that conform the GoM as

follow in L74 to L79 (In addition, the reference Tanahara (2004) was deleted from the manuscript): "Based on a detailed analysis of the central and western GoM by Portela (2018), within the gulf there are seven water masses: remnants of the Caribbean Surface Water (CSWr: also referred to as CW), North Atlantic Subtropical Underwater (NASUW), Gulf Common Water (GCW), Tropical Atlantic Central Water (TACW), the nucleus of the (TACWn), Atlantic Intermediate Water (AAIW) and North Atlantic Depth Water (NADW)"

QR2-I do not understand why the Loop Current track is called Caribbean Water. Discernment must be used throughout the MS when the authors refer to the Caribbean Water and the Loop Current; CW could be used to name the water intrusion from the Caribbean with specific biological and physical characteristics (maybe when they talk about Chl-a characteristics) and the LC when you refer to dynamic characteristics, i.e. current that enters the Yucatan Channel, loops at the eastern GoM and through the Florida Strait.

ANSWER: We agree with the comment. Following the suggestions from the referee, we define the CW in L81-82:The CW enters the gulf via the LC with specific biological (i.e low Chl-a) and physical characteristics (warmer and salty waters).

QR2-Lines 66-67: I do not consider that the MS shows a detailed analysis of the Loop Current Eddies, maybe the approach could be focus to the analysis of the LC and the LCE path footprint.

ANSWER: We rewrite in L67-69. . ."Unlike previous studies, this work entails the analysis of the LC and the LCE path footprint, and of the dominant features of the surface circulation that transports Caribbean Water (CW) into the GoM".

QR2-Line 77: Who is this acting as a primary forcing mechanism of the Loop Current? Yucatan Current? The term PFM is not clear and must be specified.

ANSWER: We deleted PFM and rewrote in L81: "The CW enters in the GoM via the

LC

QR2-Line 100: "which move CW". . .. . ..where?

ANSWER: We add in Line "move CW into the GoM"

QR2-Lines 104-105: "In this work we reexamine the effect of . . .." Rephrase please.

ANSWER: We rephrase as (L120-124): "In this work, 25-years (1993-2017) of daily ADT data are used to investigate the variability of the transport of Caribbean surface water into the gulf and its effect on chlorophyll-a concentrations using monthly radiance data from 1998-201., We examined temporal changes, mean differences, and regional concentration tendencies."

QR2-Line 121: "we considered eddies in any state of formation, detaching. . ..", then they are Loop Current Eddies (LCE), not only eddies.

ANSWER: We rephrase (L44) "In this work, we considered LCE's in any stage of formation. . ."

QR2-Lines 146-154: The first three points of the paragraph referring to data description are repetitive with the first paragraph of section 2; the description of the datasets is disordered, since they are described in two different places. I suggest to add additional information, such as the years of the data that you are using at the beginning of the section, before methods description. The calculation description of the mesoscale instabilities, as well as the AR can be described in the methods section, where they have already been described (without paragraph mark).

ANSWER: We deleted the three points of the paragraph marks. These were incorporated in the first paragraph on section 2 because of repetition with lines..... We also added the years in which the data were collected at the beginning of the section (L128-137), as follow: "Three independent data sets were used to provide evidence of temporal variability in the extension of CW into the GoM. We used the ADT and surface velocity fields (geostrophy and Ekman) from the GEKCO (Geostrophic Ekman

Current Observatory, Sudre et al., 2013) product from 1993 - 2017 with a resolution of 0.25ËŽx0.25ËŽ, in conjunction with Chl-a ocean color data derived from the reprocessing R2014.0 product suite from Aqua MODIS (Moderate Resolution Imaging Spectroradiometer) and from SeaWIFS (Sea-Viewing Wide Field of view Sensor), using the OCx Algorithm with a spatial resolution of 9X9 km (https://oceancolor.gsfc.nasa.gov/cgi/l3). The 2003-2017 monthly Chl-a ocean color product was derived from Aqua MODIS and the 1998-2002 monthly Chl-a ocean color product was derived from SeaWIFS".

QR2-For the comments "The calculation description of the mesoscale instabilities, as well as the AR can be described in the methods section, where they have already been described (without paragraph mark).

ANSWER: We removed the paragraph mark.

QR2-Line 164: Is repetitive with line 110.

ANSWER: We removed the line.

QR2-Line 192: The contours from the ADT are not determined by the influence of the CW, please rephrase.

ANSWER: We rephrase (L195-196) "In this work, we used the ADT to track both the LC and the LCE′s formed by the influence of the CW".

QR2-Please define which months of the year are described for each season (I guess winter (Jan, Feb, Mar), spring (Apr, May, Jun), etc.?).

ANSWER: We define the months of the year described for each season as follows in L204 to 207: "Fig. 1 shows that mostly in November to April, the CW retracts to its most southeasterly location. In contrast, in May to October, CW penetration moves towards the northwest. In fact, the extension begins in May and reaches maximum penetration in August, showing an annual pattern".

QR2-It is not clear to me, at least from Fig. 1, that the LC extension is retracted in autumn and extended in spring (lines 200-203), or the fact that the maximum penetration occurs in September, instead of August (see also your statement in line 232). I think it could be useful for the authors to discuss their results with previous work referring about the seasonality of the LC (see comment above).

ANSWER: We decided to remove seasons and instead refer to months for more accurate reference (See lines 204-207). This will be discuss as follow (See L207-211): Fig. 1 shows that (predominately) from November to April, the CW retracts to its most southeasterly location. In contrast, from May to October, CW penetration moves towards the northwest. In fact, this extension begins in May and reaches maximum penetration in August, resulting in an annual pattern. This movement is similar to that observed by Chang and Oey (2013). They found that in summer, the maximum LC intrusion was forced by the trade winds. Their and our observations are also consistent with the work of Candela et al (2019) who reported that water transport through the Yucatan Channel into the GoM in July was at a maximum.

QR2-Line 241 and Fig.3: Please specified if the ADT was spatially averaged.

ANSWER: In Lines 237 to 239 we added: The monthly intrusions of the CW were tracked by taking as a reference the northernmost latitudes (hereafter CWF) and westernmost longitudes of the 40 cm ADT isoline representing 1993-2017 monthly average values of the ADT (not spatially averaged).

QR2-The steric signal included in the ADT data must be discussed, considering the high-energy observed in the annual period (see Hall & Leben, 2016).

ANSWER: Lines 246 to 249 we wrote: In this work, the ADT signal also includes the seasonal steric effect, and the spectral analysis reveals a high-energy peak in the annual frequency. Base on Hall and Leben (2016) a steric signal appears as an annual sine wave with 5.8 cm amplitude, the details of which are described below.

QR2-Section 3.2 (Fig. 4): Why the STD contour of 15 cm was chosen as a reference

for the regions of maximum variability?

ANSWER: In Lines 273 to 275 we add "The STD contour of 15 cm was selected considering that this value was three times greater than the annual steric signal reported by Hall and Leben (2016)".

QR2-Lines 278-280: How this cycle of the monthly ratios compares with the results of Chang & Oey (2012, 2013)?

ANSWER: In Lines 281 to 283 we compare the results as follow: Chang and Oey (2012, 2013) proposed that the LC intrusion and the shedding of the LCE present a biannual cycle. The biannual cycle can also be related to the annual lowest and highest ratio values.

QR2-It makes not sense to me the discussion of the monthly averages of the wind stress, shown in Fig. 6 and described in section 3.4, it is not relevant for the main objective of this section, further this discussion do not reinforce the main idea exposed here; I suggest to delete this part or move this description elsewhere in the MS.

ANSWER: We agreed with the reviewer and we removed the section and Figure 6 about wind stress.

QR2-Lines 300-301: Please be more specific; Do you mean an upwelling? Is so where? Please use references.

ANSWER: The lines 300 to 301 were part of section 3.4 (Was removed).

QR2-The whole paragraph of section 3.5 is not linked with the rest of the MS, if you want to keep it, at least it go deeper in the implications and discussion of these calculations.

ANSWER: We also removed those lines because we are agree with the reviewer.

QR2-Statement of line 327-329: The difference between the mean life of the LCEs (6.8 vs 11.7) needs to be discussed with previous studies.

ANSWER: We complemented as follow in lines 313-316: "These observations also agree with the results of Lindo-Atichati et al. (2013), confirming that, on average, the LC northward intrusion starts to increase in 2002. These authors also report an increase in number/year of LC rings over the same period that also coincided with a significant increase in sea height residuals (2.78 $\pm$ 0.26 cm/decade from 1993–2009)."

QR2-Lines 378-379: How these three periods were chosen? Why not using the same two periods of the ADT? Please explain.

ANSWER: We decided to separate and evaluate the extreme period pattern from the transitional data.

QR2-Table 1 (lines 386-399), please specify what means the bold numbers in Table 1 and show the difference between Early and Contemporary periods at each row. Using the differences obtained here discuss the significance of the Chl-a averages between these two periods.

ANSWER: The bold numbers represent the average CHl-a concentrations. We add the meaning for bold numbers in table 1 as follow: "Table 1. Bold numbers denote average Chl-a concentrations (mg m-3)…….." We also include the differences between Early and Contemporary periods at each row. For discussion see section 3.5.

QR2-Second point mark of the conclusions (line 515). I do not see that in Fig. 10, or in Fig. 11.

ANSWER: We agree with the reviewer and we deleted.

Technical corrections:

QR2-Please avoid the double space throughout the manuscript (i.e., lines 58, 60, 144, 263, 267, 269, etc.).

ANSWER: Done. Spacing was corrected.

QR2-Caribbean water is mentioned for the first time in the MS in line 68 as the acronym

CW and in the line 75 as Caribbean Water. The acronyms must be defined for the first time as they are mentioned in the text and then they should be used throughout the MS as an acronym (the same for Loop Current as LC, which is even lowercase in line 99).

ANSWER: The acronyms were defined in the Introduction and used through the MS.

QR2-Line 119: Specifiy the years used for the 25 years climatologies.

ANSWER: these issue was corrected, it was written (1993 to 2017).

QR2-Line 140: 'island' instead of Island.

ANSWER: This error was corrected.

QR2-Line 216 and 230: CWF instead of Caribbean water front, it is already defined in line 125.

ANSWER: Corrected.

QR2-Lines 294-295: Please specify that this is a supposition.

ANSWER: We calculate the area and this is not a supposition.

QR2-Line 314: Please use the accurate terms. ANSWER: The section 3.5 as was mentioned before it was removed.

QR2-Line 324: 'extended to the west'. . .in summer and autumn?

ANSWER: Done.

QR2-Line 337: Avoid the use of acronyms in the title sections (especially if it has not been previously defined).

ANSWER: Corrected.

QR2-Line 360: Use CW.

ANSWER: Done.

QR2-Line 402: Add (see also Fig. 11).

ANSWER: Added.

QR2-Lines 413-415: Rephrase.

ANSWER: Done (See L371-373) as follow: This is also evident in the LC core, where Chl-a concentrations also decreased with time and signals the entrance to the gulf of more water oligotrophic during the middle and contemporary epochs".

QR2-Line 432: Check the point marks.

ANSWER: Done.

QR2-Lines 446-450: Rephrase the way you mention the M-K work please.

ANSWER: We do not understand this request.

QR2-Please avoid the use of 'we', instead use something like 'in this work. . .'

ANSWER: Because this is optional we prefer to use "We.

QR2-Line 470: R means correlation?

ANSWER: Correlation Coefficient (R)

QR2-Line 480: Change 'begs' for a more appropriate word (needs, requires, etc.).

ANSWER: it was changed for "requires"

QR2-Lines 508 and 509: Please change the term 'lifespan'.

ANSWER: time life

QR2-Line 513: Please rephrase the first point of the conclusions.

ANSWER: it was done as follow: The intrusion of the CW by LC-LCEs extends further

into the western GoM than was previously known.

---

## Author Response (AR1)

We would like to thanks the reviewer for the positive criticism and recommendations. Below you will find a point-by-point response to all comments to the both reviews. We also include a version with tracking changes.

Here the list of all relevant changes made in the manuscript:

- We removed the discussion and figure of the monthly averages of the wind stress described in section 3.4 (including figure 6), but also we removed sections 3.5 (also figure 7) and section 3.7 (AMOC).

- We rewrote all section 3.2.

- We improve the introduction and discussion regarding previous studies, which referrer to the seasonal cycle of the LC. But also the conclusion regarding the decrease of Chl-a concentrations in the western Gulf of Mexico (GoM).

- We finally highlight the relevant contributions in the MS.

**Answers to Reviewer No.1:**

I would like to thanks the reviewer for taking the effort with this paper and for his/her helpful comments. Below you find a point-by-point response to all comments. Original reviewers' comments in regular typeface, response in bold-italic letter.

1. A minor point is to what extent some aliasing may have been introduced by standardizing the satellite data into the same spatial resolution.

*R: Our major conclusion about predictable seasonality with respect to Caribbean water intrusion into the Gulf and its extent in recent years comes from ADT satellite data but without standardization. Our major conclusion about seasonality comes from the ADT satellite data before it was standardized and potentially aliased by fitting it to a standard grid.*

*In our work satellite chlorophyll data were also used. We found minimum surface Chl-a concentrations during the summer-autumn period inside the region of maximum incursion of the CW. The satellite chlorophyll data were spatially standardized from 1/12 to 1/4 degrees to obtain clean and smoothed figures (Figures 8, 9 and 10; before Figures 10, 11, 12, see figures attached to this document) without noise introduced by submesoscale activity. However, to confirm that the results do not change with or without the standardization of the data, we computed Figures 8, 9 and 10 directly with the original chlorophyll data at 1/12 degree. We did not find differences.*

2. It is not clear that the authors are keeping in mind an inherent limitation of the data in that all the data sets they are analyzing are essentially surface or near surface (a small part of the overall circulation). This is germane to the comparisons made and between Eckman and geostrophic flow regime patterns as well as other issues raised.

*R: We were very conscious of the inherent limitation of all the data sets because they were obtained from the surface or near surface. We agree the data from surface just represent a small part of the overall circulation and may lead to errors in Ekman and geostrophic flow regime patterns.*

*The vertical extent of the Ekman effect depends on the degree of wind stress and its duration. Considering this, the Ekman current impacts a layer from the surface to 30 or, unusually, up to 100 m. Geostrophic currents computed by absolute dynamic topography represent the layer from 500 m to around 1500 m, which is the mean depth of the detached eddies of the Loop Current. When considering Ekman derived geostrophic currents, we need to consider only the first 100 m. But this does not affect the results of this paper. We have removed this section.*

3. While they properly conclude that their analysis "suggests" (see section 3.6) larger volumes from 2003 onward it is by no means conclusive (see comment above).

*R: As we did not make any direct current measurements, we agree with the referee that our analysis "suggests" an increased influx of Caribbean Water has entered the GoM. See our L49-L51 in abstract section.*

4. I have issues with section 3.7 AMOC both in that they proceed as if it were shown definitively that a greater volume of Caribbean water is entering the Gulf and their use of the Caesar paper. They also then elaborate upon AMOC and synoptic scales which is pure speculation and unrelated to their own analysis.

*R: Again, we agree about the speculation and that it is unrelated to our analysis. We have removed this section.*

5. With respect to the satellite chlorophyll data the authors do not appear to understand the limits of the data. It is not only that only surface (or near surface) pigment concentration is measurable by satellite, it is more fundamentally the case that changes in the measurement can be indicative of many things other than changes in plant biomass. There is particular sensitivity to changes in plankton community structure (therefore pigment type and concentration per unit biomass). Not only are some of the differences noted smaller than I for one would be comfortable as conclusive but in fact differences in community structure in many oceanic regions (including the GoM) have been widely reported and indeed are expected given warming, acidification and changes in nutrient loading. None of this is to say that over the deeper regions of the GoM plankton biomass has not decreased but it simply cannot be rigorously inferred from this analysis.

*R: We are grateful to the referee for pointing out potential problems in establishing the relationship between upwelled radiance and biomass, and indeed this could be a source of error in coastal waters. However, we have not revised the manuscript on this issue because today chlorophyll derived from ocean color is globally accepted as the index of chlorophyll in oceanic (case 1) waters, namely oligotrophic waters such as those in the central Gulf of Mexico (GoM). Additionally, our observations of chlorophyll are supported by our independent ADT-based analysis of the annual intrusion of very low productivity Caribbean Water (CW), which shows increasing intrusion into the GoM after 2002. Finally, we have looked at the chlorophyll issue from several points of view and are confident of our conclusion. In our work we can only say that according to these satellite "products", we find a time-dependent diminution of the chlorophyll signal. This diminution has been widely*

*observed by others (Behrenfeld et al., 2006, Polovina et al., 2008; Irwin and Oliver, 2009, Laffoley & Baxter., 2016).*

[Figure]

Figure 8

[Figure]

Figure 9

[Figure]

Figure 10

**Answers to Reviewer No.2:**

- There is a lack of discussion regarding previous studies, which referrer to the seasonal cycle of the LC, like Chang & Oey (2012, 2013), which expose a bimodal cycle obtained with model outputs; Hall & Leben (2016), which show an annual cycle with altimetry data; or Candela et al. (2019), which estimate the amplitude of the seasonal cycles of the transport at Yucatan Channel and at the Strait of Florida using mooring observations. Those works, among others, must be at least mention in the introduction and their findings should be discussed with what was found in this MS.

**ANSWER: About the lack of discussion regarding previous studies that refer to the seasonal cycle we add in L97-105:**

*"LC extension and anticyclonic eddy separation are the result of the momentum imbalance (Pichevin and Nof, 1997) and form the shape of future LCEs. Chang and Oey (2010) using a numerical model, proposed that the wind stress could be the primary forcing that releases LCEs. In a second paper, supported by satellite observations, they proposed that the LC intrusion and the shedding of the LCEs followed a biannual cycle (Chang and Oey, 2013). A reanalysis of archived data also detected statistically significant LCEs separation seasonality (Hall and Leben, 2016). Recently, Candela et al. (2019) analyzed four years of water current data and reported a seasonal cycle in the transport through the Yucatan channel with the annual cycle as the main harmonic peak in July.*

- The conclusion regarding the decrease of Chl-a concentrations in the western Gulf of Mexico (GoM), where the authors state that there is a reduction due to a larger volume of waters coming from the Caribbean Sea toward the GoM, should be taken with caution and considers/discuss the limitations of the data series used for this conclusion. There are other sources that could contribute to a reduction in Chl-a in the western GoM. It is not obvious to me what the authors state with the footprint of the CW/LC path, in the Chl-a climatological patterns (Fig. 11).

**ANSWER: We agree with the reviewer about being cautious. However, without data on changes of time-dependent optical properties caused by population shifts ( ex. to picoplankton) we can only speculate. Therefore, we clarified our statement in L431 to L436.**

*"Our own results and conclusions are based on SeaWifs and AquaMODIS chlorophyll data, which in Type One water, correlate very well with chlorophyll measured with standard laboratory methods (Mati Kahru, personal communication). In our work we can only say that according to these satellite products, we find a time-dependent diminution of the Chl-a signal. This diminution has been widely observed by others although in other waters (Behrenfeld et al., 2006, Polovina et al., 2008; Irwin and Oliver, 2009, Laffoley and Baxter., 2016).*

**About the state with the footprint of the CW/LC path, in the Chl-a climatological patterns (Fig. 11) we agree and we improve the lines 324-326 as follow:**

*"During spring-summer, when the maximum CW penetration occurs, our data confirms that the footprint of the CWF water (delineated by the 40 cm isoline of ADT) is in general oligotrophic indicating that Caribbean water has indeed entered the GoM".*

- The MS need to focused to highlight the relevant contributions.

**ANSWER: We feel that the Summary and Conclusion section amply states our results and conclusions, starting with the summary of our observations on L 438-to L465:**

*"The availability of a large spatial extension of satellite observations of ADT, sea surface currents, wind stress over a quarter of century and Chl-a over 20 years has enabled us to confirm the LC and CW dynamics observed in the 60's and 70's with more recent in situ observations. The verification of the CWF climatologies developed in this work is important as a reference baseline for further numerical modeling, and it impacts assessments of the gulf's biogeochemistry, energy, heat transport, and Chl-a concentration. A recent committee of the National Academic of Sciences, (2018) suggested three main study topics to advance the knowledge of the processes that characterize the GoM: 1) the LC system active area, 2) the variation of the inflows of the LC system, and 3) the dynamic interactions of the LC system in*

*the west. Following these suggestions, we have confirmed that the maximum influence of the CW into the GoM (e,g., its maximum extension into the gulf or intrusion) has a temporal variability, being stronger in summer and weaker in the late fall and winter. This is supported by the fact that the generated monthly EKE maps have the maximum gradient at the periphery of the CWF and have a similar monthly pattern of extension and retraction as the CWF.*

*We noted that in the summer months the wind stress from the southeast is weak, thereby minimizing the flow of nutrients to the surface and causing Chl-a to be low, specifically for three reasons: 1) The increase in the surface temperature of the water column strengthens stratification 2) The intrusion of the CW to the western gulf´s surface thickens the surface layer, and 3) The eddy-driven anticyclonic circulation deepens the nutricline. This contrasts with the cold seasons, when the surface temperature of the water is lower and the northerly winds are stronger, favoring the flow of nutrients to the surface.*

*The three-year running averages of ADT 40 cm isoline reproduce qualitatively the climatological pattern of a quarter of a century showing that before 2002 the CWF was less intrusive and the LCEs sizes were smaller. In the 1993-2002 period, we calculated that the mean life cycle of the eddies was 6.8 months and that in the 2003-2015 period the mean life cycle was 11.7 months. This difference suggests that after 2003, larger volumes of oligotrophic waters from Caribbean Sea have invaded the western GoM and reduced mean surface Chl-a concentrations."*

**Additionally, we have summarized the above in three points L467 to L473.**

- Additionally, statistical relevance of the seasonal cycle needs to be deeply discussed; it is necessary to specify how representative are the climatological averages (Fig. 1), obtained from the monthly averages of the individual years (using information from Fig. 4).

**ANSWER: We improved section 3.1 by specifically naming the months and pointing out in the caption of Fig. 3 (old 4) that at the 95% confidence level, the geographical positions of the of the 40+/- 2.2 cm isolines are virtually identical.**

- Description of section 3.2 (lines 229 – 262 and lines 271 – 280) needs to be rewrite in order to guide the reader to the months of the year that the authors are talking about (maybe a Table will be useful); What is the main message given by all these numbers and description? Besides, there need to be an agreement between the months of the year with the LC extension/retraction throughout the MS.

**ANSWER: All section 3.2 was rewritten and throughout the MS we were careful to connect the months of the year to the LC extension/retraction. We have made it clear that maximum LC influx occurs in the summer months in synchrony with higher temperatures and low chlorophyll.**

**Specific comments**

- The MS focus the discussion on the oligotrophic CW intruding the GoM owing to the Loop Current. Therefore, the description of the water masses that conform the GoM, must be introduced and supported by additional bibliography, aside from Nowlin & McLellan (1967), since the reference of Schmitz (2005) is not referred to water masses composition. Further Tanahara (2004), cannot be consulted.

**ANSWER: We introduce the description of the water masses that conform the GoM as follows in L75 to L80 (In addition, the reference Tanahara (2004) was deleted from the manuscript):**

I do not understand why the Loop Current track is called Caribbean Water. Discernment must be used throughout the MS when the authors refer to the Caribbean Water and the Loop Current; CW could be used to name the water intrusion from the Caribbean with specific biological and physical characteristics (maybe when they talk about Chl-a characteristics) and the LC when you refer to dynamic characteristics, i.e. current that enters the Yucatan Channel, loops at the eastern GoM and through the Florida Strait.

**ANSWER: We agree with the comment. Following the suggestions from the referee, we**

**define the CW in L81-82:**

*CW enters the gulf via the LC with specific biological (i.e low Chl-a) and physical characteristics (warmer and saline waters).*

- Lines 66-67: I do not consider that the MS shows a detailed analysis of the Loop Current Eddies, maybe the approach could be focus to the analysis of the LC and the LCE path footprint.

**ANSWER: We rewrote L67-69…**

*"Unlike previous studies, this work entails the analysis of the Loop Current (LC) and the LC eddies (LCEs) path footprint, and of the dominant features of the surface circulation that transport Caribbean Water (CW) into the GoM …".*

- Line 77: Who is this acting as a primary forcing mechanism of the Loop Current? Yucatan Current? The term PFM is not clear and must be specified.

**ANSWER: We deleted PFM and rewrote in L81-82:** "*CW enters in the GoM via the LC.....*

- Line 100: "which move CW". . .. . ..where?

**ANSWER: We rephrased this in Line113 115.**
*Despite extensive research, after more than a half-century we are still struggling to completely understand LC variability, the processes controlling the loop current extension, and the mechanism of detachment of anticyclones from the loop.*

Lines 104-105: "In this work we reexamine the effect of . . .." Rephrase please.

**ANSWER:** *We rephrase as (L118-122):*

*"In this work, 25-years (1993-2017) of daily ADT data combined with monthly radiance data from 1998-2017 are used to investigate the variability of the transport of Caribbean surface water into the gulf and its effect on Chl-a concentration. We examined temporal changes, mean differences, and regional concentration tendencies."*

- Line 121: "we considered eddies in any state of formation, detaching. . ..", then they are Loop Current Eddies (LCE), not only eddies.

**ANSWER: In line 139 rephrase (L121) "We considered LCEs in any stage of formation…"**

- Lines 146-154: The first three points of the paragraph referring to data description are repetitive with the first paragraph of section 2; the description of the datasets is disordered, since they are described in two different places. I suggest to add additional information, such as the years of the data that you are using at the beginning of the section, before methods description. The calculation description of the mesoscale instabilities, as well as the AR can be described in the methods section, where they have already been described (without paragraph mark).

**ANSWER: We deleted the three bullets of the paragraph:. These were incorporated in the first paragraph on section 2 . We also added the years in which the data were collected at the beginning of the section (L127-133), as follow:**

*"Three independent data sets were used to provide evidence of temporal variability in the extension of CW into the GoM. We used ADT and surface velocity fields (geostrophy and Ekman) from the GEKCO (Geostrophic Ekman Current Observatory, Sudre et al., 2013) product from 1993 - 2017 with a resolution of 0.25˚x0.25˚, in conjunction with Chl-a ocean color data derived from the reprocessing R2014.0 product suite from Aqua MODIS (Moderate Resolution Imaging Spectroradiometer) and from SeaWIFS (Sea-Viewing Wide Field of view Sensor), using the OCx Algorithm with a spatial resolution of 9X9 km (https://oceancolor.gsfc.nasa.gov/cgi/l3). The 2003-2017 monthly Chl-a ocean color product was derived from Aqua MODIS and the 1998-2002 monthly Chl-a ocean color product was derived from SeaWIFS."*

- For the comments "The calculation description of the mesoscale instabilities, as well as the AR can be described in the methods section, where they have already been described

(without paragraph mark).

**ANSWER: We removed the bullet.**

- Line 164: Is repetitive with line 110.

**ANSWER: We removed the line.**

- Line 192: The contours from the ADT are not determined by the influence of the CW, please rephrase.

**ANSWER: We rephrase (L189-190) "*In this work, we used the ADT to track both the LC and the LCEs formed by the influence of the CW*".**

- Please define which months of the year are described for each season (I guess winter (Jan, Feb, Mar), spring (Apr, May, Jun), etc.?).

**ANSWER: We define the months of the year described for each season as follows in L197 to 202:**

*"Fig. 1 shows that (mostly) in autumn (October, November and December) and winter (January, February and March), the CW retracts to its most southeasterly location. In contrast, in spring (April, May, June) and summer (July, August, September), CW penetration moves towards the northwest. In fact, the extension begins in May and reaches maximum penetration in August, showing an annual pattern."*

- It is not clear to me, at least from Fig. 1, that the LC extension is retracted in autumn and extended in spring (lines 200-203), or the fact that the maximum penetration occurs in September, instead of August (see also your statement in line 232). I think it could be useful for the authors to discuss their results with previous work referring about the seasonality of the LC (see comment above).

**ANSWER: We rewrite as follows (See L197-205):**

*"Fig. 1 shows that (predominately) from November to April, the CW retracts to its most southeasterly location. In contrast, from May to October, CW penetration moves towards the*

*northwest. In fact, this extension begins in May and reaches maximum penetration in August, resulting in an annual pattern. This movement is similar to that observed by Chang and Oey (2013). They found that in summer, the maximum LC intrusion was forced by the trade winds. Their and our observations are also consistent with the work of Candela et al. (2019) who reported that water transport into the GoM in July through the Yucatan channel was at a maximum."*

- Line 241 and Fig.3: Please specified if the ADT was spatially averaged.

**ANSWER: In Lines 230 to 232 we added:**

*The monthly intrusions of the CW were tracked by taking as a reference the northernmost latitudes and westernmost longitudes of the 40 cm ADT isoline representing 1993-2017 monthly average values of the ADT (not spatially averaged).*

- The steric signal included in the ADT data must be discussed, considering the high-energy observed in the annual period (see Hall & Leben, 2016).

**ANSWER: Lines 238 to 240 we wrote:**

*In this work, the ADT signal also includes the seasonal steric effect.. Based on Hall and Leben (2016), a steric signal appears as an annual sine wave with a 5.8 cm amplitude.*

- Section 3.2 (Fig. 4): Why the STD contour of 15 cm was chosen as a reference for the regions of maximum variability?

**ANSWER*: In Lines 266 to 268 we add "The STD contour of 15 cm was selected because this value was three times greater than the annual steric signal reported by Hall and Leben (2016)".*

- Lines 278-280: How this cycle of the monthly ratios compares with the results of Chang & Oey (2012, 2013)?

**ANSWER: In Lines 273 to 275 we compare the results as follow:**

*Chang and Oey (2012, 2013) proposed that the LC intrusion and the shedding of the LCEs*

*followed a biannual cycle. The biannual cycle can also be related to the annual lowest and highest ratio values .*

- It makes not sense to me the discussion of the monthly averages of the wind stress, shown in Fig. 6 and described in section 3.4, it is not relevant for the main objective of this section, further this discussion do not reinforce de main idea exposed here; I suggest to delete this part or move this description elsewhere in the MS.

**ANSWER: We agreed with the reviewer and we removed the section and  Figure 6.**

- Lines 300-301: Please be more specific; Do you mean an upwelling? Is so where? Please use references.

**ANSWER:  The lines 300 to 301 were part of section 3.4 and were removed).**

- The whole paragraph of section 3.5 is not linked with the rest of the MS, if you want to keep it, at least it go deeper in the implications and discussion of these calculations.

**ANSWER: We also removed those lines because we are agree with the reviewer.**

- Statement of line 327-329: The difference between the mean life of the LCEs (6.8 vs 11.7) needs to be discussed with previous studies.

**ANSWER: We complemented as follow in lines 304-308:**

*"These observations also agree with the results of Lindo-Atichati et al. (2013), confirming that, on average, the LC northward intrusion starts to increase in 2002. These authors also report an increase in number/year of LC rings over the same period that also coincided with a significant increase in sea height residuals (2.78 ± 0.26 cm/decade from 1993–2009)"*

- Lines 378-379: How these three periods were chosen? Why not using the same two periods of the ADT? Please explain.

**ANSWER: We decided to separate and evaluate the extreme period pattern from the transitional data.**

- Table 1 (lines 386-399), please specify what means the bold numbers in Table 1 and show the difference between Early and Contemporary periods at each row. Using the differences obtained here discuss the significance of the Chl-a averages between these two periods.

**ANSWER***: The bold numbers represent the average CHl-a concentrations. We add the meaning for bold numbers in Table 1 as follow: "Table 1. Bold numbers denote average Chl-a concentrations (mg m$^{-3}$)......." We also include the differences and percentages between Early and Contemporary periods at each row. For discussion see section 3.5.*

- Second point mark of the conclusions (line 515). I do not see that in Fig. 10, or in Fig. 11.

**ANSWER: We agree with the reviewer and we deleted that sentence from the conclusion.**

**Technical corrections:**

Please avoid the double space throughout the manuscript (i.e., lines 58, 60, 144, 263, 267, 269, etc.).

**ANSWER: Done. Spacing was corrected.**

Caribbean water is mentioned for the first time in the MS in line 68 as the acronym CW and in the line 75 as Caribbean Water. The acronyms must be defined for the first time as they are mentioned in the text and then they should be used throughout the MS as an acronym (the same for Loop Current as LC, which is even lowercase in line 99).

**ANSWER: The acronyms were defined in the abstract and Introduction and used through the MS.**

Line 119: Specifiy the years used for the 25 years climatologies.

**ANSWER: this  issue was corrected, and we specified (1993 to 2017).**

Line 140: 'island' instead of Island.

**ANSWER: This error was corrected.**

Line 216 and 230: CWF instead of Caribbean water front, it is already defined in line 151.

**ANSWER: Corrected.**

Lines 294-295: Please specify that this is a supposition.

**ANSWER: We have reworded the phrase to say " We suppose that the greater percentage area of the STD may be attributed to a greater influence of Caribbean Sea water.**

Line 314: Please use the accurate terms.

**ANSWER: The section 3.5 was removed and sections were renumbered.**

Line 324: 'extended to the west'. . .in summer and autumn?

**ANSWER: Corrected in line 294-295.**

Line 337: Avoid the use of acronyms in the title sections (especially if it has not been previously defined).

**ANSWER: Corrected**

Line 360: Use CW.

**ANSWER: Done.**

Line 402: Add (see also Fig. 11).

**ANSWER: Added in line 356.**

Lines 413-415: Rephrase.

**ANSWER: Done (See L362-364) as follow:** *This effect is also evident in the LC core, where Chl-a concentrations decreased with time and signals the entrance to the gulf of more oligotrophic water during the middle and contemporary epochs".*

Line 432: Check the point marks.

**ANSWER: Done.**

Lines 446-450: Rephrase the way you mention the M-K work please.

**ANSWER: We repharesed M-K for  Müller-Karger** *et al.* **(2015).**

Please avoid the use of 'we', instead use something like 'in this work. . .'

**ANSWER: Because this is optional we prefer to use "We, it is more direct.**

- Line 470: R means correlation?

**ANSWER: Correlation Coefficient (R) in Line 417.**

- Line 480: Change 'begs' for a more appropriate word (needs, requires, etc.).

**ANSWER: "Begs"  was changed for "requires"**

Lines 508 and 509: Please change the term 'lifespan'.

**ANSWER:  changed to "lifetime" in line 297.**

- Line 513: Please rephrase the first point of the conclusions.

[revised manuscript text omitted]

FIGURE 5

[Figure]

FIGURE 6

[Figure]

FIGURE 7

[Figure]

FIGURE 8

[Figure]

FIGURE 9

[Figure]

FIGURE 10

[Figure]

FIGURE 11

[Figure]

---

## Author Response (AR2)

**Thank you very much Editor for the minor comments and recommendations. Below you will find our point-by-point response:**

1) Line 41 in the abstract states that wind stress was analyzed. As this analysis was taken out at the suggestion of a reviewer, "wind stress" should be removed from this line.

**R: Yes, we removed "Wind stress"**

2) Lines 81-84. These lines were added at the suggestion of a reviewer. It would be helpful to readers if the authors could add what they mean by "warmer and more saline waters." Give some data on the mean salinity and temperature of the CW and how it compares with the rest of the surface water in the Gulf of Mexico.

**R:  We add differences for salinity and temperature as follow:**

**"CW enters in the GoM via the LC with specific biological (i.e., low Chl-a) and physical characteristics (warmer by ~0.6 units and less saline waters by ~0.5 units)".**

3) Lines 264-275: this paragraph doesn't make sense as written. It includes the sentence (line 268) "From May to August, the monthly ratio descended to 1.36." However, in line 272 the August ratio is given as 1.60. Line 268 also says "Ratio values greater than 1 were found in February…" Presumably this should be greater than 1.6. Please check the numbers given for each month and the mean values for the seasonal periods. Is this really a biannual cycle? It looks as though numbers are low in summer, high in fall/winter/spring.

**R: We agree about the confusion and we recalculated the ratios (see figure below) and improved the section as follows (lines 268-275):**

Ratio values were 1.37 in January, increased to 1.45 in February, and to 1.60 in March, then peaked in April (1.63) to decrease in May (1.47) and June (1.46), reaching a low value in August (1.27). The monthly ratio then increased in September to 1.55, decreased slightly in October (153), reached a maximum value in November to 1.65, and settled to 1.55 in December. A plot of these monthly ratios clearly shows a strong biannual signal peaking in April and November (not pictured). Chang and Oey (2012, 2013) proposed that the LC intrusion and the shedding of the LCE followed a biannual cycle. This biannual cycle can also be related to the annual lowest and highest values of the ratio.

[Figure]

4) Line 393: this would be clearer if written as "be attributed to the different ways in which the two groups treated the data."

**R: We agree and we include your recommendation in line 393.**

5) Schmitz (2005) does not appear to be in the text and should be removed from the reference list.

**R: We remove Schmitz (2005)**

[revised manuscript text omitted]
 | | **0.180** (±0.047, n=4026) | **0.167** (±0.048, n=4866) | **0.173** (±0.0624, n=4828) | **0.007 (4%)** |
| Loop Current | Winter | **0.149** (±0.052, n=536) | **0.129** (±0.064, n=647) | **0.117** (±0.062, n=645) | **0.032 (21%)** |
| Western GoM | | **0.114** (±0.033, n=3693) | **0.087** (±0.049, n=4658) | **0.0834** (±0.036, n=4754) | **0.030 (27%)** |
| Loop Current | Spring | **0.0948** (±0.074, n=526) | **0.085** (±0.1287, n=642) | **0.0835** (±0.116, n=648) | **0.011 (12%)** |
| Western GoM | | **0.0887**(±0.024, n=3924) | **0.080** (±0.022, n=4794) | **0.0755** (±0.023, n=4837) | **0.013 (15%)** |
| Loop Current | Summer | **0.109** (±0.217, n=535) | **0.091** (±0.171, n=647) | **0.0938** (±0.148, n=648) | **0.015 (14%)** |
| Western GoM | | **0.151** (±0.052, n=3894) | **0.137** (±0.044, n=4876) | **0.127** (±0.043, n=4846) | **0.024 (16%)** |
| Loop Current | Autumn | **0.138** (±0.128, n=525) | **0.1325** (±0.114, n=643) | **0.122** (±0.103, n=648) | **0.016 (12%)** |

**FIGURE CAPTIONS:**

Fig. 1. Monthly means of absolute dynamic topography (ADT) and surface currents averaged over a quarter of a century (1993-2017).

Fig. 2. Climatological monthly maps of eddy kinetic energy (EKE) in GoM: red color contours correspond to the areas of maxima EKE. The heavy black line corresponds to the isoline of 40 *cm* 2.2 *cm* of the CWF (the contour of the CWF is significant at the 95% of level). The EKE was calculated using daily maps of satellite-derived currents from AVISO (GEKCO) for a quarter of a century (1993 – 2017).

Fig. 3. Geographical positions of the CWF tracked using the 40 *cm* ADT isoline representing 1993-2017 monthly average values: a) Northward and b) Westward, respectively; c) ADT spectral analysis in a region influenced by the CWF (91.25ºW, 23.125ºN and 83.5ºW, 28.12ºN).

Fig. 4. The ADT quarter-century CWF (1993-2017) monthly climatology and its standard deviation are shown in heavy and dotted lines, respectively. The heavy line corresponds to the 40 *cm* isoline of the CWF. The dotted line encloses values of the standard deviation >15 *cm*.

Fig. 5. Average monthly percentage surface areas of CW in the interior of the Gulf of Mexico determined from climatology of the STD contour > 15 *cm*; enclosed areas were calculated in relation to the GoM area ($1.56 \times 10^6$ *km²*).

Fig. 6. Monthly means of absolute dynamic topography (ADT) from 1993-2002 (color) and its respective CWF computed with the 40 *cm* isoline (heavy black line).

Fig. 7. Monthly means of absolute dynamic topography (ADT) from 2003-2017 (color) and respective CWF computed with the 40 *cm* isoline (heavy black line).

Fig. 8. Monthly climatologies of *Chl-a* (SeaWIFS, 1998-2002 and MODIS data source, 2003-2017). The heavy black line represents the contour of the 40 *cm* ADT data that represents the CWF (1998-2017). *Chl-a* values larger than 1 *mg m⁻³* are plotted in red.

Fig. 9. From top left to bottom right, average *Chl-a* values according to period: column 1, SeaWIFS 1998-2002, column 2, MODIS 2003-2008, and column 3, MODIS 2009-2014. From top to bottom figures correspond to the mean seasons. Average *Chl-a* concentration is computed inside the white and red squares (white corresponds to the western GoM and red corresponds to the LC area). Average values for each time period and season are in Table 1.

Fig. 10. Differences of *Chl-a* concentration (*mg m⁻³*) between 2009-2014 average values of MODIS data minus 1998-2002 average SeaWIFS values. The broken line represents the 250 *m* isobath. White contoured areas indicate no significant differences.

Fig. 11. *Chl-a* concentrations (*mg m⁻³*) at four stations (a to d) in the GoM, daily time series derived from SeaWIFS from 1998 to 2002 (green) and MODIS from 2003 to 2017 (blue). Least square regressions for SeaWIFS (red line), MODIS (cyan line), and the overall linear regressions for each station (dashed black line).

FIGURE 1

FIGURE 2

[Figure]

EKE (cm$^2$s$^{-2}$)

[Figure]

FIGURE 4

[Figure]

[Figure]

FIGURE 6

[Figure]

FIGURE 7

[Figure]

[Figure]

Chlorophyll-a (mg m$^{-3}$)

FIGURE 9

[Figure]

FIGURE 10

[Figure]

FIGURE 11

[Figure]